# TOPMELT 1.0: A topography-based distribution function approach to snowmelt simulation for hydrological modelling at basin scale

Mattia Zaramella[1], Marco Borga[1], Davide Zoccatelli[1], and Luca Carturan[1,2]

[1]Department of Land, Environment, Agriculture and Forestry, University of Padua, Padova, 35020, Italy
[2]Department of Geosciences, University of Padua, Padova, 35131, Italy

**Correspondence:** Mattia Zaramella (mattia.zaramella@unipd.it)

**Abstract.** Enhanced temperature-index distributed models for snowpack simulation, incorporating air temperature and a term for clear sky potential solar radiation, are increasingly used to simulate the spatial variability of the snow water equivalent. This paper presents a new snowpack model (termed TOPMELT) which integrates an enhanced temperature index model into th ICHYMOD semi-distributed basin scale hydrological model by exploiting a statistical representation of the distribution of clear sky potential solar radiation. This is obtained by discretising the full spatial distribution of clear sky potential solar radiation into a number of radiation classes. The computation required to generate a spatially distributed water equivalent reduces to a single calculation for each radiation class. This turns into a potentially significant advantage when parameter sensitivity and uncertainty estimation procedures are carried out. The radiation index may be also averaged in time over given time periods. Thus, the model resembles a classical temperature-index model when only one radiation class for each elevation band and a temporal aggregation of one year is used, whereas it approximates a fully distributed model by increasing the number of the radiation classes and decreasing the temporal aggregation. TOPMELT is integrated within the semi-distributed ICHYMOD model and it is applied at hourly time step over the Aurino basin at S. Giorgio, a 614 km$^2$ catchment in the Upper Adige river basin (Eastern Alps, Italy) to examine the sensitivity of the snowpack and runoff model results to the spatial and temporal aggregation of the radiation fluxes. It is shown that the spatial simulation of the snow water equivalent is strongly affected by the aggregation scales. However, limited degradation of the snow simulations is achieved when using ten radiation classes and four weeks as spatial and temporal aggregation scales, respectively. Results highlight that the effects of space-time aggregation of the solar radiation patterns on the runoff response are scale dependent. They are minimal at the scale of the whole Aurino basin, while considerable impact is seen at a basin scale of 5 km$^2$.

## 1 Introduction

Seasonal snow cover is an important storage and source of melt water for human use, irrigation and hydropower production in many regions of the world. On the other hand, snow cover and melt water can be a cause of disastrous natural hazards, such as floods and avalanches. Additionally, snow cover is a key factor in the weather and climate system, both regionally and globally (Armstrong et al. , 2008). Owing to the society's strong need for updated information on snow conditions, snow accumulation and melt, models have been developed with a wide range of features. Approaches for snowpack computation

range from empirical models (e.g. simple temperature index models) to more-sophisticated physically based energy-balance models (Zappa et al., 2003; Vionnet et al., 2012; Essery et al., 2013; Magnusson et al., 2015; Avanzi et al. , 2016). Temperature-index models require only temperature as input and are based on the assumption of a linear relationship between this variable and melt rates, whereas energy-balance models are based on the computation of all relevant energy fluxes at the snowpack

surface, and thus require extrapolation of numerous meteorological and surface input variables at local scale (Jóhannesson et al., 1995; Cazorzi and Dalla Fontana, 1996; Hock, 1999; Hock and Holmgren, 2005; Anslow et al. , 2008; Formetta et al., 2014). Advances over the simple dependence of melt on air temperature by addition of radiation terms have been suggested in the last decades (Hock, 1999; Pellicciotti et al., 2005; Carturan et al., 2012), and are termed enhanced temperature-index models (ETI) here. In contrast to simple temperature-index models, where melt varies in space only as a function of elevation

(given by temperature lapse rates), ETI models includes a term for clear sky potential solar radiation. This term accounts for topographic effects (e.g. aspect, slope and shading) on the spatial distribution of melt, without the need for additional meteorological variables (e.g. global radiation and cloud data). ETI models have been found to provide a better representation of the spatial and temporal variability of melt controlled by solar radiation, when compared with simple temperature index models, as reported by a number of authors (Cazorzi and Dalla Fontana, 1996; Hock, 1999; Pellicciotti et al., 2005; Carenzo et

al., 2009 among others). Some of these approaches also better cope with the physical character of the melt process and provide a promising approach to modelling the snowpack at the catchment scale with fewer input data than energy-balance models, but allowing better model parameter transferability than standard temperature-index models (Carenzo et al., 2009).

In spite of their improved accuracy compared to simpler approaches, ETI models have been so far not integrated within lumped or semi-distributed, basin-scale hydrological modelling schemes, which are still frequently used to model sparsely

gauged mountainous catchments. Integration of ETI snowpack models into lumped or semi-distributed hydrological models may have the potential to increase spatial transferability of calibrated snowpack model parameters for hydrological applications over ungauged mountainous basins, as shown by Comola et al. (2015). Another important implication of the stronger physical basis of the ETI model with respect to simpler 'degree day' models is that it might be more appropriate for the study of climate-change impact on melt regimes, as shown by Pellicciotti et al. (2005). Finally, increasing the accuracy of the modelled snow

water equivalent may improve the outcomes of data assimilation procedures of remotely sensed snow cover information. In a few cases, semi-distributed models incorporating a spatial discretization in classes of elevation and aspect have been applied to represent the effect of exposition on snow melt and adjust snowpack parameters accordingly (Klok et al., 2001; Konz and Seibert, 2010; Abudu et al. , 2016). In other cases, a mean value of radiation has been used over the basin area in the same elevation band, modifying accordingly the melt parameters (Li and Williams, 2008). However, these types of tessellation have

no allowance for representing the actual variations of radiation distribution over space and time, which is an important feature in ETI models (Pellicciotti et al., 2005).

This work describes a novel snowpack model (termed TOPMELT herewith), which integrates the ETI snowpack method originally developed in a spatially distributed way by Cazorzi and Dalla Fontana (1996) within a semi-distributed basin-scale hydrological model. In the model developed by Cazorzi and Dalla Fontana (1996), local snowmelt is computed by using a

combined melt factor which is multiplied by a radiation index and positive air temperature. With TOPMELT, pixels with

similar radiation index and air temperature are identified by subdividing basin elevation bands into a number of radiation index classes. Then, the snowpack modelling is carried out for each class of radiation index and for each elevation band. This ensures to achieve the significant computational efficiency, which characterizes the temperature index models, allowing at the same time the stronger physical basis of ETI models. This is a potentially significant advantage when several model simulation runs

should be carried out, such as in Monte Carlo based parameter sensitivity and uncertainty estimation procedures.

The model accounts for the temporal variability of the radiation index by using local mean values of the index computed over given temporal aggregation intervals, ranging from one to several weeks . This means that a time-averaged solar radiation distribution is used over a given temporal interval, before substituting it with a new averaged distribution. With decreasing the updating interval, the accuracy of the model is expected to increase at the expenses of the computational efficiency.

As the spatial distribution of clear sky solar radiation changes with time, a radiation class computed over two different periods may sample two different portions of the elevation band. This means that a pixel belonging to a certain class at a given time, will belong to a different class at another time. TOPMELT incorporates a time-integration routine, which accounts for the temporal variability of the radiation index distribution, ensuring a consistent temporal simulation of the snowpack. Thus, TOPMELT permits full implementation of the ETI snowpack method taking into account the seasonal evolution of the spatial

distribution of solar radiation. Moreover, it provides a spatially continuous mapping of simulated snow water equivalent, in spite of the computationally-efficient semi-distributed representation of basin-scale snowpack modelling. Depending on the number of radiation classes which are used in the model, the snowpack model makes use of solar radiation values which are spatially averaged over different areas. Effectively, the model resembles a classical temperature-index model when only one radiation class for each elevation band is used, whereas it approximates a fully distributed model with increasing the number

of the radiation classes (and correspondingly decreasing the area corresponding to each class).

This paper describes in detail the structure of TOPMELT and of the time-integration routine. The integration of TOPMELT within the ICHYMOD hydrological model is also illustrated. Finally, results are reported from the application of TOPMELT over the $614 \, \text{km}^2$ Aurino basin at S. Giorgio in the Upper Adige river system (Eastern Italian Alps). The case study is exploited to i) examine the sensitivity of the snowpack and runoff model results to the temporal and spatial aggregation of the radiation

fluxes, and ii) to identify suitable spatial and temporal aggregation intervals for model simulation. The sensitivity analysis is performed on modelled snowpack in terms of snow water equivalent and on ensueing simulated runoff, comparing the output from simulations performed at different aggregation intervals with a reference represented by the finest aggregation levels.

## 2   TOPMELT structure

In TOPMELT, the basin area is subdivided into elevation bands to account for air temperature variability with elevation. Then,

each elevation band is subdivided into a number of radiation classes. This is carried out by dividing each elevation band into a number $n_c$ of equally distributed radiation classes, where the *i-th* class contains the band sub-area corresponding to the *i-th* percentile of the incident radiation energy. Therefore, the model spatial domain is represented by $n_b$ elevation bands and by $n_c$ radiation classes for each elevation band. TOPMELT deals with separate snow and glacier melt: to account for the presence of

a glacier area associated to an energy class, each one of the $n_b \times n_c$ model cells is characterized by the corresponding fraction of glacier area. The spatial subdivisions controls the balance between computational efficiency and model accuracy in the snowpack model.

The following sections describe the main input and modules of the model, where an hourly temporal interval is used for model computations.

## 2.1 Clear sky potential radiation computation and derivation of radiation distributions

For the application of TOPMELT presented in this work, clear sky short wave solar radiation [Wm$^{-2}$] is computed at each element of the Digital Terrain Model (DTM) by taking into account shadow and complex topography, calculating the apparent sun motion (Swift , 1976; Lee, 1978; Oke, 1992) and the intersection of radiation with topography (Dubayah et al., 1990; Ranzi and Rosso, 1991). Diffuse radiation is computed by accounting for self-shading (by slope and aspect) and occlusions produced by the visible horizon. Since the model uses radiation values averaged over a given time interval, maps of potential radiation averaged over time are also computed. The spatial distribution of time-averaged clear sky solar radiation are calculated over each elevation band, and $n_c$ equally distributed radiation classes are identified. For each radiation class, the mean daily cumulated clear sky radiation value is computed (termed Radiation Index RI herewith, [MJ m$^{-2}$]) and used in the snowmelt computation. Radiation is pre-processed and provided as model input. Therefore, the relative module is not included in TOPMELT code, but made available as a stand-alone tool (see the code availability section at the end of the paper).

## 2.2 Computation of precipitation amount and phase

Snow accumulation is computed starting from estimates of precipitation and air temperature, based on air temperature and precipitation data from the available weather stations. Similarly to radiation, the model permits use of several techniques, ranging from Thiessen's polygons to multi-quadratic (Borga and Vizzaccaro, 1997) for the estimation of basin mean areal precipitation values, which are provided to the model as input data. For the analyses reported in this work, the Thiessen method was used to calculate the mean precipitation over the basin. Air temperature data are used to estimate an unique hourly vertical lapse rate for the whole basin.

To account for gauge catch deficiencies that occur during periods of snow, precipitation data are corrected with a Snow Correction Factor ($SCF$). This is a multiplier of the precipitation data which is applied when station temperature is lower than a threshold temperature $T_c$. Finally, the basin precipitation value $p_{basin}$ is obtained by applying a non-dimensional Precipitation Correction Factor ($PCF$) to account for poor spatial representativeness of rain-gauge stations.

TOPMELT computes the precipitation value at the *i-th* elevation band $p_i$ [mm h$^{-1}$] by using a vertical precipitation gradient, accounting for increased precipitation over elevation. Based on results from Tuo et al. (2016), this is obtained by means of a precipitation gradient $G$ [km$^{-1}$], as follows:

$$p_i = p_{basin} \cdot \left( 1 + G \cdot \frac{h_i - h_{ref}}{1000} \right) \tag{1}$$

where $h_i$ and $h_{ref}$ [m a.s.l.] are the mean altitude of the *i-th* elevation band and of the basin respectively. Eq. 1 is applied in a way to modify only the distribution of precipitation across the elevation bands without altering the value of $p_{basin}$.

$PCF$ and $G$ parameters are generally obtained by comparing model-based snow cover simulations with satellite-based snow-cover estimates. Since the average areal precipitation $p_{basin}$ is a TOPMELT input, only $G$ is a TOPMELT parameter. Optionally, TOPMELT permits to provide a different precipitation value precipitation for each elevation band.

Temperature $T_i$ [°C] is provided as input for each elevation band and time step. In this work, a mean value of air temperature $T_i$ over the *i-th* elevation band is obtained by using the aforementioned vertical temperature lapse rate. Estimation of precipitation phase (solid or liquid) is therefore performed over each elevation band, according to the threshold temperature $T_c$.

## 2.3 Computation of snow and ice melt

For the generic model cell represented by the *i-th* elevation band and the *j-th* radiation class, snow melt rate $f_{i,j}(t)$ [mm h$^{-1}$] at time $t$, is computed taking into account air temperature, clear sky radiation and albedo. During day hours, the snowmelt is given by:

$$f_{i,j}(t) = CMF \cdot RI_{i,j}(t) \cdot d \cdot (1 - ALB_i(t)) \cdot \max\left[0, (T_i(t) - T_b)\right] \qquad (2)$$

where: $T_i(t)$ is the elevation band temperature, $RI_{i,j}(t)$ [MJ m$^{-2}$] is the cell radiation index; $d$ [-] is the fraction of daylight hours in a day (i.e. the number of daylight hours divided by 24); CMF [mm °C$^{-1}$MJ$^{-1}$m$^2$h$^{-1}$] is the combined melt factor, accounting for both thermal and radiative effects; $alb_i(t)$ [-] is the albedo of snow, $T_b = 0$ °C is a threshold base temperature. Snow albedo is computed for each elevation band based on Brock et al. (2000):

$$alb_i(t) = ALBS - \beta_2 \cdot \left[ \log_{10} \sum_k T_i(t_k) \right] \qquad (3)$$

where $ALBS$ [-] is the fresh snow albedo, $\beta_2$ [-] is a dimensionless parameter, $\sum_k T_i(t_k)$ [°C] is the sum of the positive hourly temperatures exceeding the threshold base temperature $T_b$ since the last snowfall until the current time $t$.

During night hours, snow melt is simulated accounting only for air temperature, as follows:

$$f_{i,j}(t) = NMF \cdot \max\left[0, (T_i(t) - T_b)\right] \qquad (4)$$

where $NMF$ [mm h$^{-1}$ °C$^{-1}$] is the Night Melt Factor.

For rain-on-snow conditions (Anderson , 1976), melting is computed depending on air temperature and on the energy provided by rain:

$$f_{i,j}(t) = \left[ RMF + \frac{p_{i,j}(t)}{COST} \right] \cdot \max\left[0, (T_i(t) - T_b)\right] \qquad (5)$$

where $RMF$ [mm h$^{-1}$ °C$^{-1}$] is the Rain Melt Factor and $COST$ [°C$^{-1}$] is a parameter accounting for the influence of rain on snowmelt (Carturan et al., 2012). For each model cell, the snow water equivalent ($we_{i,j}$ [mm]) is updated by accounting for

snow accumulation, rain-on-snow, melt and freezing water. Water due to snowmelt or rainfall is first retained in the snowpack as interstitial water termed liquid water $liqw_{i,j}$ [mm]. When Liquid Water exceeds a water holding capacity of the snowpack (termed $LWT$), this propagates through the snowpack at a rate $DYTIME$ [m h$^{-1}$], to form net water flow at the snowpack base.

When air temperature is less than the threshold base temperature, part of the liquid water refreezes and $liqw_{i,j}$ is reduced and added to the snowpack through a freezing rate, termed $ice$ [mm h$^{-1}$]. This is computed as:

$$ice_i(t) = REFRZ \cdot \min\left[0, (T_b - T_i(t))\right] \tag{6}$$

where $T_b$ is the threshold base temperature (Eq. 2) and $REFRZ$ [mm °C$^{-1}$ h$^{-1}$] is the freezing factor. When $we_{i,j}$ is less than a threshold (termed $WETH$), ice melt starts. This is computed similarly to snow (Eq. 2), but where the snow albedo is replaced

by a constant glacial albedo, $ALBG$ [-], as follows:

$$f_{i,j}(t) = CMF \cdot RI_{i,j} \cdot d \cdot (1 - ALBG) \cdot \max\left[0, (T_i(t) - T_b)\right] \tag{7}$$

During rainfalls or night hours, glacier melt is computed by means of Eq. 4 and Eq. 5 respectively.

All model parameters and their values are listed in table 1, along with variable names and units. TOPMELT is majorly sensitive to the melt factor $CMF$, the most significant calibration parameter of the snowmelt model. It is constant both in time and

space, as the variability of the melt rate is accounted for by the radiation index and by air temperature. Other important parameters are: the fresh snow albedo, $ALBS$; the rain melt factor, $RMF$, which is a constant calculated from a simplified energy budget (Anderson , 1976); the precipitation gradient $G$, which drives the re-distribution of precipitation with the elevation; the base temperature $T_b$, which defines the fusion temperature of snow and ice.

## 2.4   Updating the Radiation Index distribution: the time-integration routine

As reported in the previous sections, the spatial distribution of clear sky solar radiation changes with time, based both on astronomic variation of the radiation flux and its interaction with a complex topography. This implies that the statistical distribution of the radiation index over each elevation band will be also modified and should be updated. In general a radiation class, computed at two different time steps, covers two different areas of the elevation band. Thus, a pixel belonging to a certain class at a given time will belong to a different class at another time. Figure 1 shows two different maps of $RI$, (representing January

1 and April 1, from an elevation band ranging from 2000 to 2200 m a.s.l., taken from the basin selected for the case study of this work. The radiation index is distributed using ten equally-distributed classes. While the radiation index varies from 1.2 to 20.5 MJ m$^{-2}$ in January, it ranges from 7.9 and 34.7 MJ m$^{-2}$ in April. However, differences are not restricted to the magnitude of the index. Despite each class has the same area within the elevation band, their spatial distribution changes from one map to the other. Examination of the figures shows that a number of pixel belonging to Class V in January are included in Class IX in

April. Since the two snowpack state variables, $we(t)$ and $liqw(t)$, are computed at the model cell level, pixel transition from a given cell to another must be accounted for, whenever the radiation index distribution is updated with time (termed switch date here). To account for pixel transition through classes, TOPMELT implements an adjusting procedure for model state variables.

**Table 1.** Model parameters and variables: short name, description and measuring units. Parameters are written with capital letters, variables in lowercase.

| Parameter | Description | Value | Units |
|-----------|-------------|-------|-------|
| $ALBG$ | Glacier albedo | 0.3 | - |
| $ALBS$ | Fresh snow albedo | 0.9 | - |
| $\beta_2$ | Dimensionless parameter for $alb$ computation | 0.0919 | - |
| $CMF$ | Combined Melt Factor | 0.013 | mm °C$^{-1}$MJ$^{-1}$m$^2$h$^{-1}$ |
| $DYTIME$ | Speed of water propagation through snowpack | 3 | m h$^{-1}$ |
| $G$ | Precipitation gradient | 0 | km$^{-1}$ |
| $LWT$ | Water holding capacity, fraction of w.e. | 0.1 | - |
| $NMF$ | Night Melt Factor | 0.16 | mm °C$^{-1}$h$^{-1}$ |
| $REFRZ$ | Freezing factor | 0.03 | mm °C$^{-1}$h$^{-1}$ |
| $RI$ | Radiation Index, mean daily energy per unit surface | $1 \div 42$ | MJ m$^{-2}$ |
| $RMF$ | Rain Melt Factor | 0.3 | mm °C$^{-1}$h$^{-1}$ |
| $T_b$ | Base temperature | 0.0 | °C |
| $T_c$ | Snow/rain threshold temperature | 1.5 | °C |
| $WETH$ | Water equivalent minimum threshold before ice-melt | 5 | mm |

| Variable | Description | | Units |
|----------|-------------|--|-------|
| $alb$ | Snow albedo (accounting for aging) | | - |
| $h$ | Elevation | | m a.s.l. |
| $f$ | Fusion rate | | mm h$^{-1}$ |
| $ice$ | Freezed water | | mm |
| $liqw$ | Interstitial melt water | | mm |
| $p$ | Precipitation rate | | mm h$^{-1}$ |
| $T$ | Temperature | | °C |
| $we$ | Water Equivalent (w.e.) | | mm |

The procedure is here described applied to the arbitrary cell state variable $x_{i,j}$, corresponding to the *i-th* elevation band and the *j-th* radiation class of the basin. When updating from one radiation index map to another, pixels from a certain class can, in principle, move to all other classes, and pixels from other classes can conversely move to that class. The $x_{i,j}$ variable, which corresponds to certain model cell, should be updated accordingly. Therefore, a 2-D array accounting for pixels transition and

5    the associated variables among classes is defined, namely the transition matrix $\mathbf{M}_i$ of the *i-th* elevation band.

    The transition matrix is $n_c \times n_c$ sized and is computed for each elevation band and is unique for each switch date of the radiation index maps. The element $M_{i,j,k}$ of the matrix represents the number of pixels moving at a switch date from the *j-th*

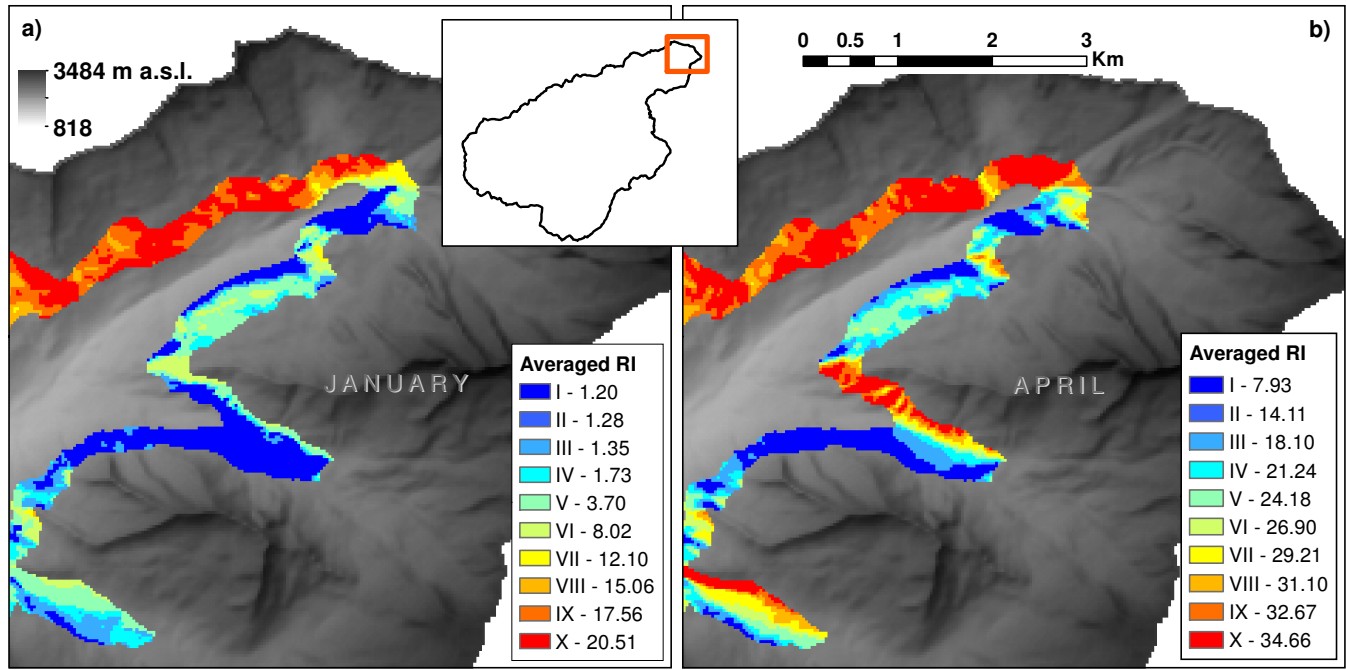

**Figure 1.** Comparison between radiation index distribution over the 2000-2200 m elevation band of the Aurino basin for a) January 1st and b) April 1st (ten classes subdivision). The figures show the north-eastern portion of the basin and report the average radiation index M[J m$^{-2}$], with the corresponding radiation class identified by a roman number.

source class to the *k-th* destination class, within a given elevation band i. $M_{i,j,j}$ is the diagonal element of $\mathbf{M}_i$, representing the pixels that did not move from the source radiation class. Provided that the total number of pixels belonging to a class must remain constant, a property of the transition matrix is that the sum of the elements along the *i-th* row is equal to the sum of the elements of the *j-th* column (i.e. the number of pixels leaving a class is replaced by an equivalent number of pixels migrating from other classes):

$$\sum_{h=1}^{n_c} M_{i,j,h} = \sum_{h=1}^{n_c} M_{i,h,j} = N_{i,j} \tag{8}$$

where $N_{i,j}$ is the number of pixels in the *j-th* radiation class within the *i-th* elevation band. When TOPMELT switches from one radiation index map to another, the cell state variable $x_{i,j}$ in the new model cell will be the sum of the pixel contribution from other classes and of the pixels remaining in the source class, where the number of incoming or remaining pixels is weighted with respect to the total number of pixels of the source class. Therefore, $x_{i,j}$ is corrected through a matrix $\mathbf{C}_i$ of correction coefficients, relative to the *i-th* elevation band, which can be derived from $\mathbf{M}_i$ through the following relation:

$$C_{i,j,k} = \frac{M_{i,j,k}}{N_{i,j}} \tag{9}$$

Follows from Eq. 8 and Eq. 9 that the sum of each line or column of the correction factors matrix $\mathbf{C}_i$ must be equal to 1:

$$\sum_{j=1}^{n_c} C_{i,j,k} = \sum_{j=1}^{n_c} \frac{M_{i,j,k}}{N_{i,j}} = 1 \tag{10}$$

The coefficient $C_{i,j,k}$ represents the correction factor for the state variable $x_{i,j}$ that must be redistributed among the other classes through the updating process, within the *i-th* elevation band. With the updating, if $x_{i,j}$ is the source class variable band and $x_{i,j}$ its transformed (i.e. destination), the class variable correction is computed through the following:

$$\widehat{\mathbf{x}}_i = \mathbf{C}_i \cdot \mathbf{x}_i \tag{11}$$

or through the equivalent forms:

$$\begin{bmatrix} \widehat{x}_{i,1} \\ \widehat{x}_{i,2} \\ \vdots \\ \widehat{x}_{i,n_c} \end{bmatrix} = \begin{bmatrix} C_{i,1,1} & C_{i,1,2} & \cdots & C_{i,1,n_c} \\ C_{i,2,1} & C_{i,2,2} & \cdots & C_{i,2,n_c} \\ \vdots & \vdots & \ddots & \vdots \\ C_{i,n_c,1} & C_{i,n_c,2} & \cdots & C_{i,n_c,n_c} \end{bmatrix} \cdot \begin{bmatrix} x_{i,1} \\ x_{i,2} \\ \vdots \\ x_{i,n_c} \end{bmatrix} \tag{12}$$

and

$$\widehat{x}_{i,j} = \sum_{k=1}^{n_c} C_{i,j,k} x_{i,k} \tag{13}$$

Eq. 13 represents the weighted sum of $x_{i,k}$ pixels that moved from the $n_c$ source classes to destination *k-th* class. Since the correction factors matrix $\mathbf{C}_i$ can be computed once for all, the model computational efficiency is preserved.

To exemplify the computational flow and its constraints, the example of the water equivalent $we$ state variable is reported here. At a given radiation index switch, $we_{i,j}$, will be transferred within the *i-th* elevation band across different classes transforming into $\widehat{we}_{i,j}$, for $j = 1, n_c$. The total volume of snow at a given elevation band i of the destination distribution is:

$$\widehat{V}_i^{we} = \sum_{j=1}^{n_c} (\widehat{we}_{i,j} N_{i,j} A_p) \tag{14}$$

where $A_p$ is the pixel size. Combining Eq. 13 and Eq. 14, and provided that the number of pixels is the same for each class of the *i-th* elevation band ($N_{i,j} = N_i$ for $j = 1, n_c$):

$$\widehat{V}_i^{we} = N_i A_p \sum_{j=1}^{n_c} \left[ \sum_{k=1}^{n_c} (C_{i,j,k} we_{i,k}) \right] = N_i A_p \sum_{k=1}^{n_c} \left( we_{i,k} \sum_{j=1}^{n_c} C_{i,j,k} \right) \tag{15}$$

Eq. 10 plus Eq. 15 yield that the transformed w.e. volume is equal to original volume:

$$\widehat{V}_i^{we} = N_i A_p \sum_{k=1}^{n_c} we_{i,k} = V_i^{we} \tag{16}$$

Therefore, Eq. 10 is a constraint that holds conservation of w.e. through the updating process.

## 2.5 Representation of the water equivalent distribution and snow cover

The model allows to provide the representation of spatially continuous water equivalent maps (as well as any other model cell variable) at a given time. This is carried out by exploiting a routine which links each model cell to the corresponding topographic elements, accounting for variation of the radiation index maps. Then, the water equivalent maps may be converted to snow cover maps by using suitable threshold values. In this work, we used a minimum threshold of 10 mm (Parajka and Blöschl, 2008) for the intercomparison with the MODIS snow cover products.

## 3 TOPMELT integration into ICHYMOD

TOPMELT is integrated within a semi-distributed hydrological model, ICHYMOD, (Norbiato et al., 2008), which transforms excess precipitation plus melt contribution into runoff at the outlet of the basin. The total flow routed from TOPMELT to ICHYMOD is the areal weighted sum of each single cell flow, which is made by rainwater and excess snowmelt water. The model consists of a soil moisture routine and a flow routing routine.

The soil moisture routine uses a probability distribution to describe the spatial variation of water storage capacity across a basin, accordingly with the Probability Distributed Model (PDM) by Moore (2007). Drainage from the soil enters slow response pathways. The base discharge is routed from groundwater to the catchment outlet through a cubic law storage model. Direct runoff from the proportion of the basin where storage capacity has been exceeded is routed by means of a geomorphology-based distributed unit hydrograph (Da Ros and Borga , 1997), conceptualized by a cascade of two linear reservoirs in series. Runoff from ice melt is transferred to the outlet through two different routes, depending on glacial till imperviousness. Part of the ice meltwater is input to the soil moisture storage, while the remaining fraction flows directly to the outlet as a cascade of two linear reservoirs in series. The base discharge is routed from groundwater to the catchment outlet through a cubic law storage model. Storage-based representations of the fast and slow response pathways yield a spatially lumped representation fast and slow response at the basin outlet which, when summed, gives the total basin flow.

Losses due to evapotranspiration are calculated as a function of potential evapotranspiration and the status of the soil moisture store in the PDM. Potential evapotranspiration is estimated by using the Hargreaves method (Hargreaves and Samani, 1982).

## 4 TOPMELT: Impact of spatial and temporal aggregation scale

### 4.1 Study site, available data and model set up

TOPMELT is applied to the Aurino river basin closed at San Giorgio, located in the Adige river system in the Eastern Alps, Italy (Figure 2). The basin has an area of 614 km$^2$, 2.7% of which covered by glaciers for a total of 16.4 km$^2$. Forest covers 33.5% of the basin, pasture and grassland 44.5%, bare soil and rocks 21.6% and urban area the remaining 0.4%. Elevation ranges from 817 to 3485 m a.s.l.. Mean annual basin averaged precipitation is around 950 mm, with values ranging from 850 mm at lower elevations to 1300 mm at the highest elevations. Precipitation and temperature data at hourly time intervals are

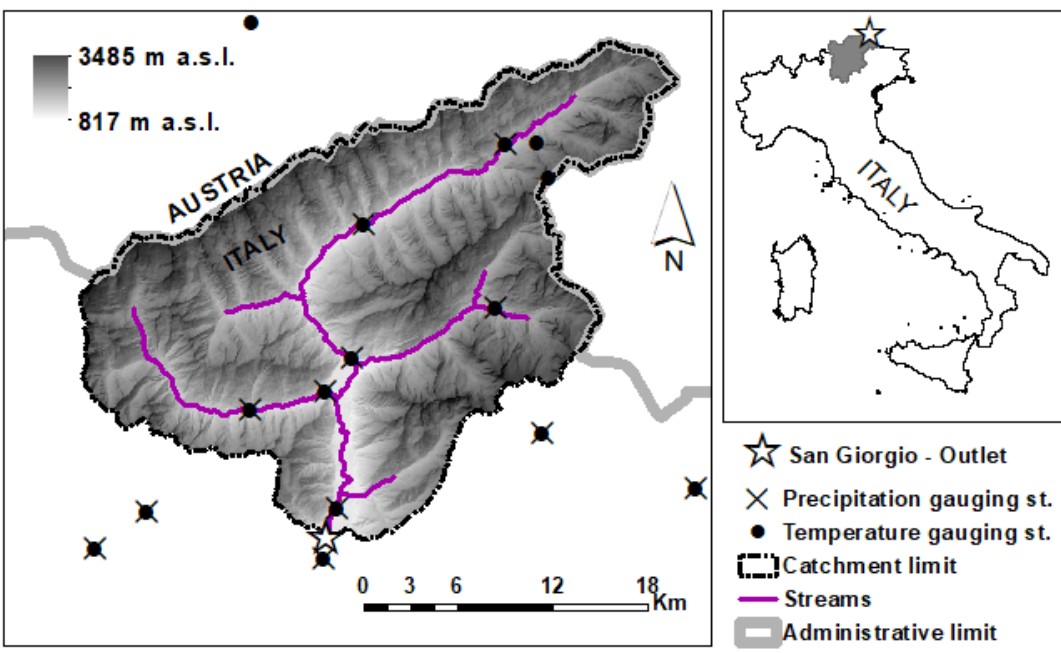

**Figure 2.** The Aurino river basin closed at S. Giorgio with the position of the hydro-meteorological monitoring stations.

provided by 15 gauging stations (see Figure 2 for locations), whereas observed discharge are available at the stream-gauge station in S. Giorgio Aurino. The natural runoff regime is partially altered by the reservoir operations over the 25 km² Neves basin. Basin topography is described by means of a DTM with a 30 m grid resolution.

Satellite observations of snow cover at 250 m resolution are available for the study basin since January 2011 and are provided
5   by an algorithm based on MODIS observations developed by Notarnicola et al. (2013a and 2013b). With this algorithm, MODIS maps provide for each pixel the presence or absence of snow, the presence of clouds, water bodies or pixels with no feature detected. 50 MODIS maps are available during the period from January 1 to June 30, 2011 with a percentage of cloud cover less than 10%.

The basin was subdivided into 14 elevation bands of 200 m each, ranging from 800 to 3600 m a.s.l.. Elevations bands were
10   then subdivided in a number $n_c$ of radiation classes. To assess the impact of different spatial aggregation levels of the radiation index on model results, five types of class subdivisions of the basin were considered. The elevation bands were divided into $n_c$=1, 5, 10, 15 and 20 classes, yielding five different spatial aggregation labelled with C1, C5, C10, C15 and C20 respectively. Similarly, to analyse the influence of using aggregation periods of the radiation index, five different updating times were used for the computation of the radiation index distribution, with duration of 1, 2, 4, 8 and 12 weeks and labelled W1, W2, W4,
15   W8 and W12 respectively. Variable temporal and spatial discretization allows for different configurations of the space-time aggregation of the radiation index. For example, label W4-C10 refers to a model set up with a temporal aggregation interval of

weeks combined with use of 10 radiation classes per elevation band. It is interesting to observe that the model set up W12-C1 resembles a traditional temperature-index model with a radiation correction for elevation band (as in Li and Williams, 2008), whereas the model set-up W1-C20 approximates a fully spatially distributed implementation of the enhanced temperature index model. One should bear in mind that when just one radiation class is used, there is no need to update the distribution of the
snow water equivalent.

### 4.2    The time integration routine: assessment of pixel transition

An important feature of the model is the use of the time-integration routine to ensure consistency in the snowpack simulation. This routine accounts for pixel transition from one radiation index class to another at the switching time. In this section we analyse the pixel transition, by using an index which represents the percentage of migrating pixels over the total number of
pixels belonging to a given elevation band, as follows:

$$MI_i = \frac{\widehat{N}_i}{N_i} \tag{17}$$

where $MI_i$ is the migration index of the transition, $\widehat{N}_i$ is the number of pixels changing class during a switch, and $N_i$ is the number of pixels of the *i-th* elevation band.

The percentage of migrating pixels was computed at four elevation bands: the lowest, from the lowest elevation of the
basin, 817 m, to 1000 m; two intermediate bands, from 1600 to 1800 m and from 2400 to 2600 m; the highest, from 3400 to 3485 m (which is the max elevation in the basin). The analysis was performed for the five temporal aggregation by using ten radiation index classes, reporting the mean average migration index over the various switches. Results are reported in Figure 3a, showing that the percentage of migrating pixels ranges from up to 16% at W1 temporal aggregation to up to 69% at W12 temporal aggregation, with a considerable increase of the transition percentage with the increase of temporal aggregation. It
is interesting to observe that the transition percentage decreases with increasing elevation of the band, i.e. with decreasing the spatial dispersion of pixels corresponding to a certain class. It is worth noting that the migration index increases with the number of radiation classes (results not shown here for brevity).

Finally, a specific analysis aimed to analyse the magnitude of the transition class change. To highlight this aspect, the percentage of pixel which moves by only one class (for example from the second to the third radiation class) was computed
and compared to the transition percentage. The percentages were computed and averaged for all the elevation bands. Figure 3b shows this comparison, by considering ten radiation classes, for the five temporal aggregations. For aggregation W12, 42% of the pixels migrates through one class, 17% through more than one class and only 41% do not migrate at all. For aggregation W1, 12% of the pixels migrates through one class, only 1% through more than one class and 87% of the pixels do not move from the source class at the switch time. These results agree with those reported in Figure 3a and underline the impact of using
larger temporal aggregations on the pixel transition between various classes.

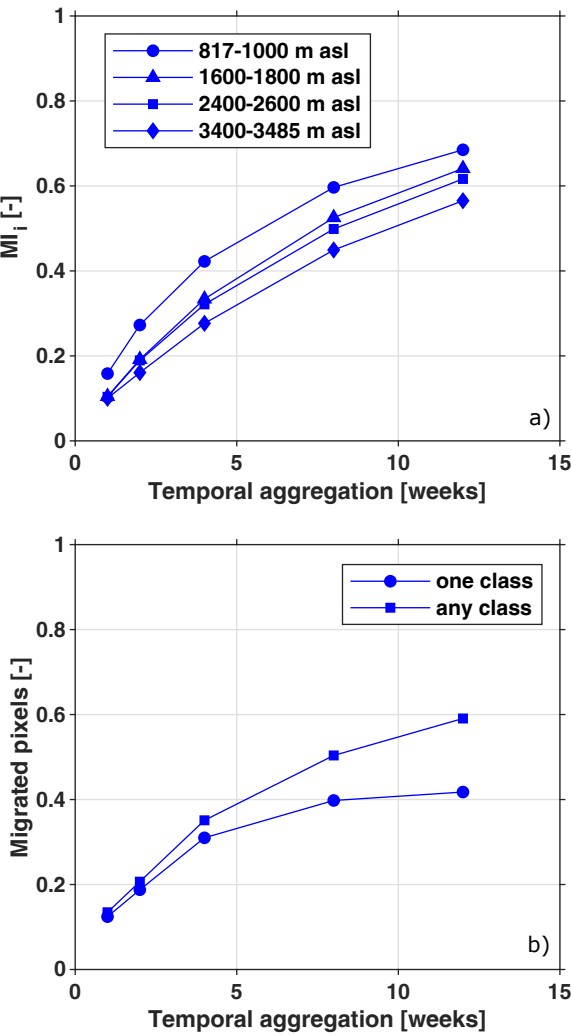

**Figure 3.** a) Band migration index for the five temporal aggregations, reported for four elevation bands (lowest elevation band from 817 to 1000 m); from 1600 to 1800 m; from 2400 to 2600 m; highest elevation band from 3400 to 3485 m). b) Fraction of migrated pixels computed for the five temporal aggregations over all the elevation bands. Circles refers to pixels that migrated of ±1 energy class, squares to total migrated pixels.

## 4.3  Model calibration and validation

The results reported in Section 4.2 are obviously independent on the specification of the snowpack model parameters. However, their impact on model results (for instance, on the snow water equivalent spatial distribution) depends on the specification of model parameters.

TOPMELT and ICHYMOD parameters were identified by means of a two-stage procedure, based on comparison of the simulated outflow with the discharge measured at San Giorgio and on comparison of the simulated snow cover with MODIS data, for the period where MODIS data were available. The model set-up W4-C10 was used for the parameter identification. The following statistics were used for comparing simulated and observed discharges:

$$BIAS = \frac{\sum_{t=1}^{N}(Q_{sim,t} - Q_{obs,t})}{\sum_{t=1}^{N} Q_{obs,t}} \tag{18}$$

$$NSE = 1 - \frac{\sum_{t=1}^{N}(Q_{sim,t} - Q_{obs,t})^2}{\sum_{t=1}^{N}(Q_{obs,t} - \bar{Q}_{obs})^2} \tag{19}$$

where $Q_{obs,t}$ and $Q_{sim,t}$ are the observed and simulated discharge at time $t$, respectively, $\bar{Q}_{obs}$ is the average value of the observed discharges, and $N$ is the number of observations. Optimal values for BIAS and NSE are 0 and 1, respectively.

To compare simulated snow cover (SC) area with MODIS observation, a snow water equivalent threshold of 10 mm was used to declare a snow covered pixel (Parajka and Blöschl, 2008). Then, the 30 m grids contributing to one MODIS pixel were calculated, and a simulated MODIS-like pixel was considered as snow covered if the percentage of the snow covered 30 m grid size pixels is equal or higher than 50%. MODIS maps with cloud coverage less than 10% were used for the analysis. For assessing the correspondence of simulated versus observed values, the Accuracy Index – ACC skill measure, based on the contingency table, was used:

$$ACC = \frac{TP + TN}{TP + FN + FP + TN} \tag{20}$$

where $TP$ are the number of true positives, i.e. where both model and observation agree on the presence of snow on the pixel; $TN$ is the number of true negatives, $FN$ is the number of false negatives. i.e. pixels which are snow covered according to MODIS and where the model simulates no snow, $FP$ is the number of false positives, i.e. pixels which are free of snow according to MODIS and where the model simulates snow. ACC ranges between 0 and 1 with its optimum at 1. Application of a comparison between MODIS data and TOPMELT simulated snow cover is exemplified in Figure 4 for a sample date: May 6, 2011. Following Parajka and Blöschl (2008), the Accuracy Index (Eq. 20) was computed on a pixel base over the 50 cloud-free MODIS maps available. The resulting spatial distribution of the Accuracy Index is termed Overall Accuracy (OA) map.

The model parameter identification was carried out by using data from October 1, 2001, to September 30, 2012 by using the model configuration W4-C10. The period from October 1, 2007 to September 30, 2012 was used for model parameter calibration, with optimization of the statistics BIAS, NSE and ACC, whereas model validation was carried out over the period October 1, 2001, to September 30, 2007. Model parameter optimization started from the parameterization of the ICHYMOD application by Norbiato et al. (2009). Model error statistics NSE and BIAS are equal to 0.71 and 2%, respectively, for the calibration period, and to 0.71 and -9%, respectively, for the validation period.

A graphical comparison of simulated (W4C10) and observed discharges is reported in Figure 5 for the period May-July 2011, showing the general consistency of the simulation.

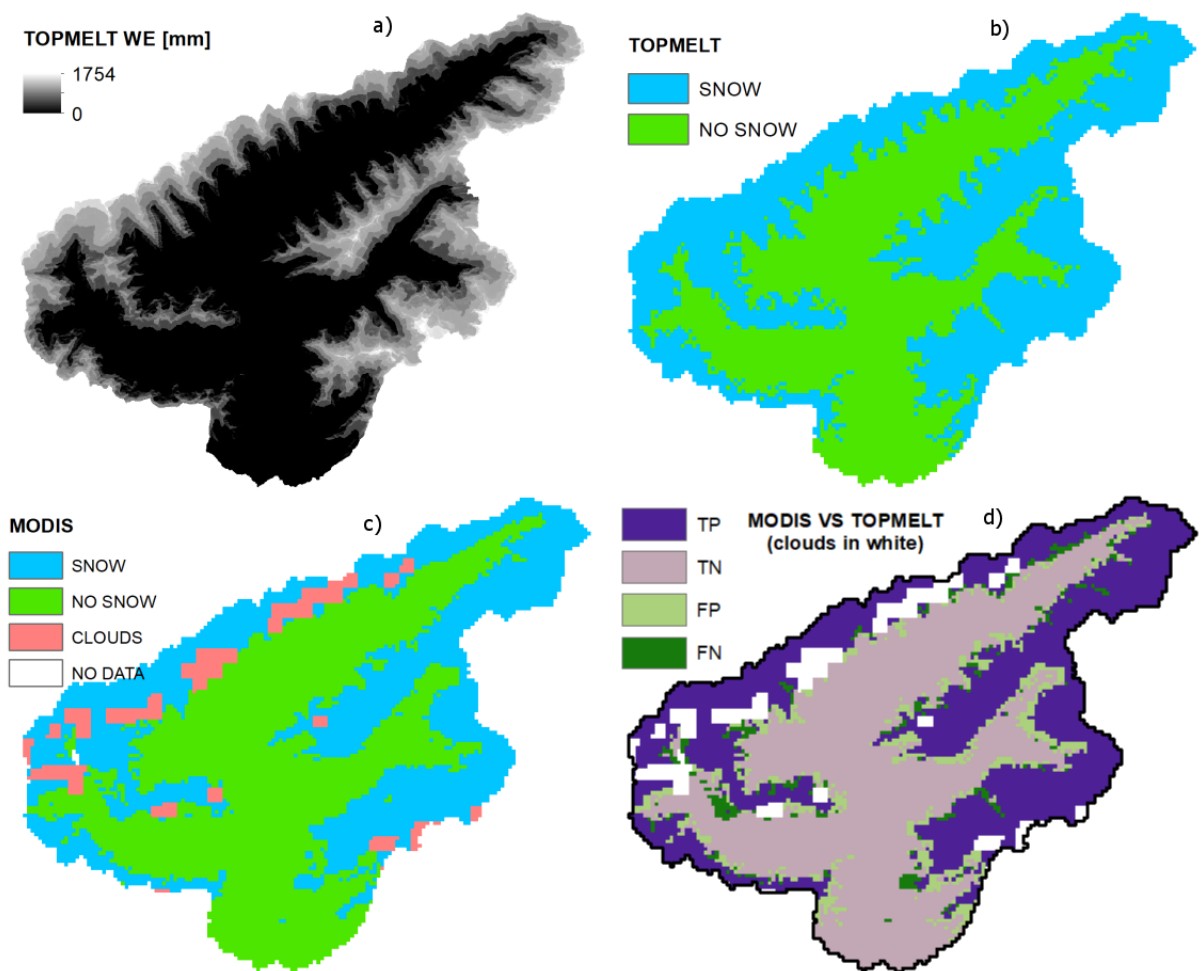

**Figure 4.** Comparison between simulated and MODIS-derived snow cover map. The comparison is obtained by using the model set up W4-C10 for May 6, 2011. TP nd TN are true positives and true negatives: both TOPMELT and MODIS are indicating the presence or absence of snow at the pixel, respectively. Similarly FP and FN are false positives and false negatives.

The Overall Accuracy map is reported in Figure 6b, while Figure 6b shows the land use over the catchment. The figure shows that simulated snow dynamics agrees (OA > 0.7) with MODIS snow cover detection by 71% of the area. Lower OA corresponds to forest cover, and north facing slope with forest cover are characterised by very low values of OA. This shows clearly the well-known combined effect of view geometry and forest cover on MODIS snow cover accuracy. Forests make MODIS remote sensing of snow challenging because the presence of trees complicates monitoring of snow using remote sensing as trees obscure snow on the ground surface (Notarnicola et al., 2013b). View geometry may be a further major error sources in MODIS snow mapping algorithms in forested areas. This is because the gaps in forest canopies, which are essentially the detectable snow fraction in winter, are lower at off-nadir views (Notarnicola et al., 2013b).

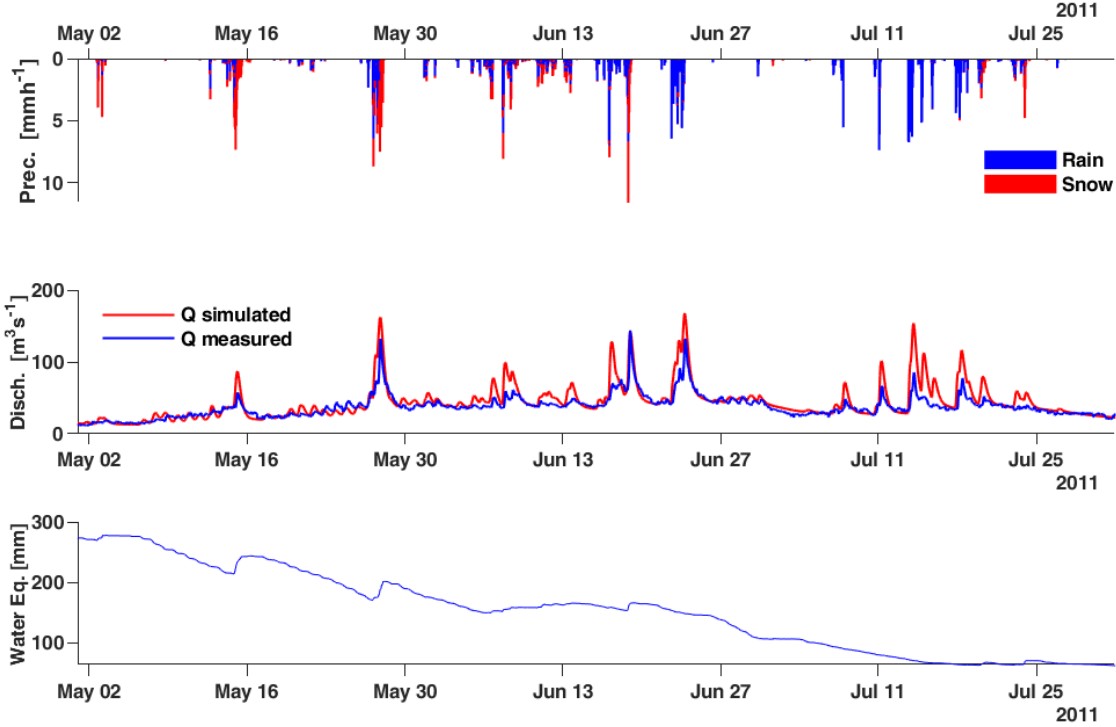

**Figure 5.** Comparison between simulated (W4-C10) and observed discharge at San Giorgio for the May-July 2011 period. The bottom plot displays the average w.e. over the basin.

## 4.4 Impact of temporal and spatial aggregation on model results

The impact of using different spatial and temporal aggregation on TOPMELT results was carried out by considering both the spatial distribution of the water equivalent and the simulated flow. In this analysis, the finest spatial and temporal discretizations, C20 and W1 respectively, were taken as a reference for the the comparisons. For the water equivalent spatial distribution, the assessment was carried out for the period between October 1, 2010 and June 30, 2011, generating a distribution of w.e. at weekly time step, for a total of 50 simulated snow maps. The intermediate subdivision into radiation classes (C10) was used for the comparison of temporal aggregations W12 to W2 with W1; on the other hand, the intermediate temporal aggregation of 4 weeks (W4) was used for the comparison of spatial aggregations C1 to C15 with C20 (i.e. the reference configuration for Figure 7a is W1-C10, for Figure 7b is W4C20).

The NSE statistic (Eq. 19) was used to quantify the agreement bewteen the analysis and reference w.e. distribution over space, by excluding from the comparison all the occurrences of zero water equivalent on both maps. One value of the NSE statistic was obtained for each of the 50 maps.

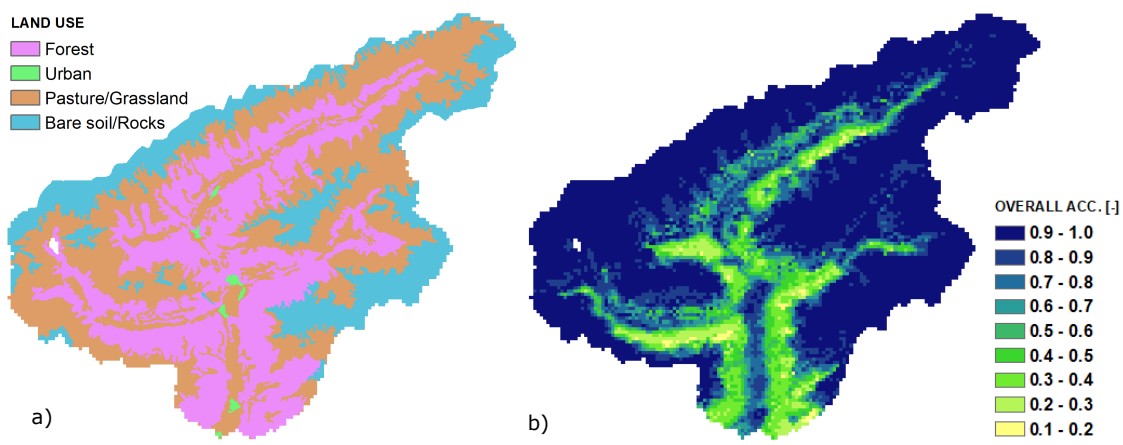

**Figure 6.** a) Land use distribution for the catchment and b) pixel based overall accuracy (OA) of the comparison between simulated and MODIS-derived snow cover maps, computed from January 1 to June 30, 2011 for a total of 50 MODIS snow cover maps.

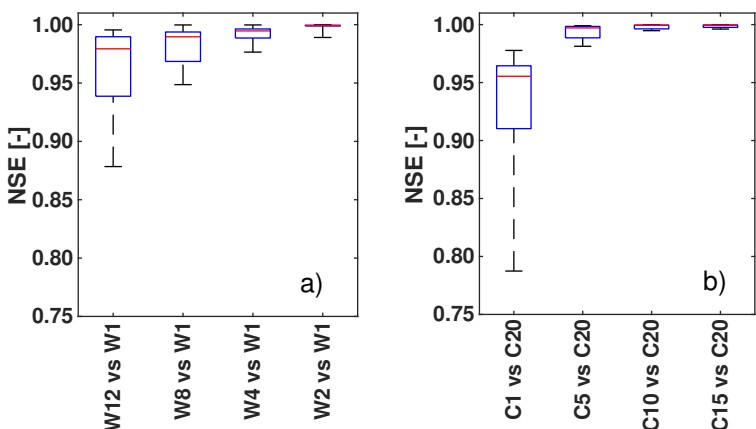

**Figure 7.** Box plots of NSE computed from the pixel by pixel comparison of the a) temporal and b) spatial aggregation series of w.e. maps, from October 2010 to June 30 2011 at weekly time-step. On each box, the central mark indicates the median, and the bottom and top edges of the box indicate the 25th and 75th percentiles, respectively. The whiskers extend to the maximum and the minimum efficiency. The reference is TOPMELT with W1-C10 configuration for the plot of the left panel, W4-C20 for the plot on the right.

The results are reported in Figure 7, where the distributions of the NSE statistics are summarised by using box plots. Figure 7a summarises the results concerning the impact of the temporal aggregation, whereas Figure 7b shows the analysis concerning the spatial aggregation. Using the longest temporal aggregation (W12) has a noticeabe impact on the results, with NSE ranging from 0.88 to 0.99. Even in this case, a considerable improvement is obtained by using a reduced temporal aggregation, such as W4. Interestingly, the results resemble the findings obtained by comparing the model set up in terms of pixel migration.

Using just one radiation class (C1) degrades markedly the accuracy of the WE distribution, with NSE ranging from 0.78 to 0.98. Results improve considerably by using 5 classes (C5).

The impact analysis on flows simulations was carried out by comparing observed and simulated flows over the validation period from October 1, 2001 to September 30, 2007. Results are reported in Table 2 by using the NS efficiency for the various spatial and temporal aggregations. The results are at odd with those reported for the w.e. distribution, showing that the sensitivity of the modelled runoff is negligible. The reported NSE values are very close to each other, ranging from 0.70 to 0.73. The comparison of each runoff simulation to the simulations obtained by using the CW reference model, as we did previously for the snow water equivalent maps, results in NSE values always in excess of 0.99. The comparison statistics varied only in a limited way when the comparison was focused on the March. 1 to June, 30 period, i.e. the melting season.

These results are not unexpected. The size of the basin, which largely exceeds the correlation scale of the radiation index, together with the branching nature of the river network, provides indeed a powerful way in averaging out the heterogeneity of snowmelt processes, as shown by Comola et al. (2015), among others.

To better highlight the control exerted by the catchment size on runoff simulations, we subdivided the study basin into a number of sub-basins characterised by different drainage areas. We isolated 5 basins with mean drainage of 20 km$^2$, 10 basins with mean drainage area of 10 km$^2$, and 20 basins with mean drainage area of 5 km$^2$. The basins were identified to ensure the sampling of the whole range of solar radiation characteristics of the main basin. The model was implemented to these basins by modifying only those model parameters which depend explicitly on the area and the topography of the basin, including, of course, the distribution of the radiation index. The other model parameters were transposed from those identified for the basin closed at S. Giorgio Aurino. In these comparisons, we analysed only the effect of varying the number of radiation classes. We considered only three subdivisions for the radiation classes: 1, 10 and 20 classes, and we used the runoff obtained by using the finest spatial resolution (20 classes) as a reference against which the other two were constrasted. We used a period of four weeks (W4) as the temporal aggregation. As a matter of fact, the simulations from the model set-up W4C1 and W4C10 were compared with the corresponding simulations from the model set-up W4C20. The NSE statistics were computed only over the March. 1 to June, 30 period, i.e. the melting season. The results, reported in Table 3 as an average value of the NSE statistic over the different basins, show clearly the control exerted by the catchment size on the effect of using a different number of radiation classes for runoff simulations. The differences between W4C1 and W4C20 runoff simulations are considerable (NSE=0.77) for 5 km$^2$ area and rapidly decrease with increasing the drainage area. At 10 km$^2$ and at 20 km$^2$ NSE amounts to 0.91 and to 0.99, respectively. On the other hand, no degradation is reported for runoff simulated by using 10 radiation classes instead of 20, at least for drainage areas equal or exceeding 5 km$^2$.

## 5   Conclusions

This paper presents TOPMELT, a parsimonious snowpack simulation model which integrates an enhanced temperature index model into a semi-distributed basin scale hydrological model. This is obtained by discretising the full spatial distribution of clear sky potential solar radiation into a number of radiation classes in each elevation band. Snowpack suimulation is carried

**Table 2.** Nash-Sutcliffe index ($NSE$) of the TOPMELT-ICHYMOD model at different spatial aggregation and temporal resolution, from October 2001 to October 2007.

| W4C1 | W4C5 | W4C10 | W4C15 | W4C20 |
|------|------|-------|-------|-------|
| 0.73 | 0.73 | 0.71 | 0.73 | 0.73 |

| W1C10 | W2C10 | W4C10 | W8C10 | W12C10 |
|-------|-------|-------|-------|--------|
| 0.71 | 0.71 | 0.71 | 0.70 | 0.71 |

**Table 3.** Mean value of the Nash-Sutcliffe index (NSE) of the comparison between W4C1 and W4C10 TOPMELT-ICHYMOD simulated flows and the reference flow simulations, obtained by using the W4C20 set up, over basins of three different drainage areas: 5, 10 and 20 km$^2$. Comparisons carried out over the March, 1 to June, 30 period.

| Model set-up | Sub-basin area | | |
|--------------|----------------|---------------|---------------|
|              | 5 km$^2$ | 10 km$^2$ | 20 km$^2$ |
| W4C1 | 0.77 | 0.91 | 0.99 |
| W4C10 | 0.97 | 0.99 | 0.99 |

out for each radiation class, rather than for each DTM pixel. This allows to develop synthesis in modelling approaches to snow simulation and provides the potential for analysing the impact of spatial and temporal aggregation of radiation fluxes on model results. Furthermore, this approach reduces computational burden, which is a key potential advantage when parameter sensitivity and uncertainty estimation procedures are carried out.

The impact of temporal and spatial aggregation of the radiation fluxes on model results is assessed by applying TOPMELT on the 614 km$^2$ Aurino river basin at S. Giorgio, in the Eastern Italian Alps. The analysis is carried out by examining five temporal aggregation levels (ranging from one to twelve weeks) and five spatial aggregation levels (obtained by subdividing each elevation band into a number of radiation classes ranging from one to twenty), with their impact on the prediction of snow water equivalent distribution and runoff response.

The assessment of the snow water equivalent simulations clearly shows the degradation of model results when using large temporal and spatial aggregation scales, with a model efficiency decreasing up to 20%. On the other hand, the sensitivity analysis shows that averaging out the radiation index over four weeks and using a ten radiation classes subdivision, has a minimal impact on model results.

Analysis of the runoff response simulations shows that the effects of the spatial patterns of snow water equivalent are strongly smoothed at the scale of Aurino basin at S. Giorgio, with minimal deviations over the different considered model set-up. Examination of TOPMELT-driven runoff results obtained over internal sub-basins ranging in area from 5 to 20 km$^2$ highlight that the effects of space-time aggregation of the solar radiation patterns on the runoff response are scale dependent, and that scale dependency is controlled by the spatial aggregation of radiation index. Simulations obtained by using just one radiation class degrade in accuracy when the model is applied over basins of around 5 km$^2$, whereas runoff simulations obtained

with using ten radiation classes show very limited degration over all the basin areas considered. These results are important to drive implementation of TOPMELT for operational applications. They may prove to be relevant for data-assimilation of remotely-sensed snow cover information, which may be made more effective with incresing the accuracy of the modelled snow cover. They may prove to be relevant also to use the spatial transferability of enhanced temperature-index model parameters to ungauged basins.

One important implication of our results concerns the transferability of the simpler temperature-index model, which is simulated in our cases when TOPMELT is implemented by using just one radiation class. The results suggest that this spatial aggregation of the radiaton patterns do not impair the spatial transferability of temperature-index models for runoff simulations of basins larger than a certain threshold, equal to 5 km$^2$ in our case study.

*Code availability.* TOPMELT is developed in Python, version 3.6, and additionally tested with Python 2.7 (Python Software Foundation, https://www.python.org/). Python installation requires the following additional modules: datetime, inspect, math, os, sys, pyodbc. The code requires the installation of a SQL database (DB) to store input data and to collects output. TOPMELT was developed and tested with MySQL Community Server (GPL), version 5.7.21. TOPMELT source code and a quick user guide are available at the repository http://doi.org/10.5281/zenodo.1342731.

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
