# Peer review of "TOPMELT 1.0: A topography-based distribution function approach to snowmelt simulation for hydrological modelling at basin scale"

_Geoscientific Model Development, 2018_

## Referee Comment (RC1) · Massimiliano Zappa (Referee) · 15 Apr 2019

Dear authors,

I evaluated your manuscript and I include a commented version of it.

I summarize here my major concerns:

LUMPED? SEMI DISTRIBUTED? Here is the definition of lumped very broad and actually the implementation with elevation bands and radiation index classes heavily reminds me the definition of hydrological response units, also a semi-distributed approach. I think your approach is much more semi-distributed than lumped.

INPUT PRECIPITATION Please expand on the techniques declared at page 4.

LIST OF VARIABLES I would welcome a Table with a list of the used abbreviations.

"DYNAMIC" RADIATION AREA AND INDEX: If you had static radiation regions instead of radiation classes you would not need the supplementary workaround for updating the states with a "migration". Can you better justify your choice, or, even better, compare you results to a version with static radiation sub areas selected using elevation, aspect and/or slope?

Best regards

Massimiliano Zappa

Please also note the supplement to this comment:
https://www.geosci-model-dev-discuss.net/gmd-2018-202/gmd-2018-202-RC1-supplement.pdf

―――――――――――――

**Supplement:**

[revised manuscript text omitted]

---

## Short Comment (SC1) · 19 Apr 2019

We would like to thank the Reviewer Massimiliano Zappa for his review of the manuscript. We provide here a prompt reply to underline some aspects concerning the modelling methodology. We hope this will facilitate further interaction on the points listed in his review. The Reviewer underlines four main points, as follows.

1. "LUMPED or SEMI DISTRIBUTED. Here is the definition of lumped very broad and actually the implementation with elevation bands and radiation index classes heavily reminds me the definition of hydrological response units, also a semi-distributed approach. I think your approach is much more semi-distributed than lumped.

[Figure]

In the manuscript, we characterised TOPMELT as 'lumped' in order to make very clear the differences with a spatially distributed approach. However, we agree with the Reviewer that the model can be considered as 'semi-distributed', as this class of models do not make calculations for every point in the catchment but for a distribution function of characteristics. TOPMELT has the feature that the snow predictions can be mapped back into space for comparison with any observations of the snow properties.

2. INPUT PRECIPITATION Please expand on the techniques declared at page 4.

With the sentence at line 13, page 4, we mean that precipitation can be calculated by using a range of methods (Thiessen Polygons, multi-quadratic and Kriging), on the condition that the model input is an areal precipitation estimated starting from any of these techniques. The interpolation and averaging code of precipitation is not included in the version of TOPMELT illustrated in this manuscript, but it is included in the complete hydrological model code. For the case study of the paper, we used the Thiessen polygons to calculate an areal precipitation over the whole basin. We will clarify this issue in the revised version of the paper.

3. LIST OF VARIABLES I would welcome a Table with a list of the used abbreviations.

We will report the list of model variables in the revised version of the paper.

4. "DYNAMIC" RADIATION AREA AND INDEX: If you had static radiation regions instead of radiation classes you would not need the supplementary workaround for updating the states with a "migration". Can you better justify your choice, or, even better, compare your results to a version with static radiation sub areas selected using elevation, aspect and/or slope?

TOPMELT accounts for the seasonality of sun declination and for the visible horizon, therefore including both the effects of the temporal variability of the incident radiation angle and of shadowing. This makes the spatial distribution of radiation variable over time, which requires the updating of the snow states and represents a key feature of

the model. We note that using of a static radiation distribution would require the identification of a reference date for which the distribution will be computed. This would bring some degree of arbitrariness in the modelling methodology. The paper already includes an analysis of the impact of decreasing the frequency of the snow state updating. This can be seen in Fig. 8, left panel, which shows the impact of updating the radiation distribution at decreasing frequency. The frequency ranges from 1 week, which is chosen as the reference temporal aggregation, to 2 weeks, 4 weeks, 8 weeks and 12 weeks. Fig. 1 below shows the scatter plots corresponding to the pixel-by-pixel comparison summarised in Fig. 8 of the paper, in terms of snow water equivalent (SWE). The scatter reported in Fig. 1 indicates that the impact of the decreasing frequency may have important consequences when the SWE spatial distribution is sought. In the revised version, we will further reduce the updating frequency in order to more completely answer the question raised by the Reviewer.
* * *
[Figure]

Fig. 1: Scatter plot of the pixel-by-pixel comparison of SWE, obtained by updating the SWE classes at decreasing frequency ranging from 2 weeks to 12 weeks. The updating frequency of 1 week is used as reference. The study period is from October 2010 to June 30 2011.

**Fig. 1.**

---

## Referee Comment (RC2) · Anonymous Referee #2 · 21 Apr 2019

General comment

In this manuscript, the authors presented a new model simulating snowpack and snowmelt (TOPMELT), integrated in a lumped basin scale hydrological model. TOPMELT is based on an enhanced temperature index accounting for elevation bands and radiation classes (obtained by clear sky radiation values derived by the analysis of Digital Terrain Models). After the presentation of the model structure, the authors presented an application of the model to a catchment in the Eastern Italian Alps for which there are long-term discharge data and snow-covered-area data derived by MODIS maps. Specifically, the authors compared snow-covered areas obtained by TOPMELT

and MODIS and analyzed the sensitivity of the model results to the spatial and temporal aggregation of the radiation data.

I think the topic of the manuscript is interesting for the readers of Geoscientific Model Development. The structure of the model is well described and clear, and the manuscript is well organized. However, I have few specific comments that should be considered by the authors and various minor corrections to the text (small errors and typos).

All the specific and minor comments are reported in the supplementary pdf file.

Specific comments

Page 4, line 14: The Precipitation Correction Factor is mentioned only here without further explanation or details. Is it an important parameter in the model? How is it determined? What are the typical values? In the revised manuscript, I suggest the authors to integrate the description of the Precipitation Correction Factor.

Page 8, line 17: How does the conversion snow-water equivalent maps to snow cover maps work? What are the 'suitable threshold values'? I suggest the authors to integrate the description in the text.

Page 9, line 9: The authors should report the estimated fractions of catchment area for the different land uses. Does the model consider interception by various vegetation covers? If so, how? How does the snowmelt module work in forested areas and are there any differences compared to other land uses?

Page 13, lines 5-8: I think coniferous forests and discontinuous vegetation might be the dominant land covers in such a catchment. Therefore, given the limits of MODIS in vegetated areas, I suggest to show the comparison between TOPMELT and MODIS for all the land covers, except for the forested areas. As an alternative, the results of the comparison could be shown in a table or in the figures, but distinguishing the various land uses.

Please also note the supplement to this comment:
https://www.geosci-model-dev-discuss.net/gmd-2018-202/gmd-2018-202-RC2-supplement.pdf

**Supplement:**

[revised manuscript text omitted]

---

## Referee Comment (RC3) · Anonymous Referee #3 · 3 May 2019

This manuscript presents enhanced a snowmelt model by combining temperature, clear sky potential solar radiation, and topographic parameters. This is very important in hydrologic model as there are few data available in the high mountains. However, this manuscript needs a significant improvement.

Major comments:

1. The structure could be improved. 1) in the Introduction, the manuscript should emphasize the gap and importance of the research and how you are going to achieve, instead of model introduction as in lines 21-35 (page 2) and lines 1-10 (page 3).2) Section 4 kind of mixes the methodology and model results

2. The model calibration is too short. For new model introduction, people are interested in model parameters, their sensitivities, or how to estimate these parameters. It is not clear here.

3. Comparison results are also missing. For example, there is only one simulation presented in Figure 6, while there are so many simulations have been done (if I am right). How does snow parameters affect hydrograph or snow cover? (see also point 2 above).

4. I am not clear how radiation is calculated. Is it based on station data or theoretical solar radiation equation? How did MODIS data come into play? Section 2.1 and/or section 4.1 could explain something on this.

5. The integration of TOPMELT and ICHYMOD is also not clear, especially on the routing. My understanding is that ICHYMOD is a lumped model and its routing scheme shouldn't consider elevation bands. Then how is water from each cell (combination of elevation bands and radiation classes) routed to the outlet?

6. English is readable. However a native speaker might improve the manuscript

7. Section 5 seems more like a summary

Minor comments:

1. Line 24 of page 1 and Line 1 of page 2. I don't understand the logic

2. There is a duplication (line 29 of page 3, and line 1 of page 4)

3. Equation 1.What is the range of G

4. Figure 6: add the content of bottom plot in the caption.

5. Figure 7: why don't you use the whole simulation period?

6. "reference fields" in the second last line of Page 14: what are they?

7. Figure 8: What is your point? To me, models with similar spatial or temporal resolution should give similar results.

---

## Author Comment (AC1) · 9 Jul 2019

Our point-to-point response is reported below. Reviewer's Comments are reproduced in italics; the Authors' Responses are given directly afterward. All reviewer comments are identified using the code RXCY, where X is the reviewer number and Y is the reviewer comment number (for example R1C3 means Reviewer 1 Comment 3). Line numbers in authors' responses refer to the original manuscript unless otherwise stated.

Additionally, we enclosed a marked-up version of the revised manuscript.

**Answers to main comments**

[Figure]

*R2C1: Page 4, line 14: The Precipitation Correction Factor is mentioned only here without further explanation or details. Is it an important parameter in the model? How is it determined? What are the typical values? In the revised manuscript, I suggest the authors to integrate the description of the Precipitation Correction Factor.*

The Precipitation Correction Factor (PCF) accounts for poor estimation of the precipitation due to climatic non-representativeness of the gauging stations. Its value may be estimated by optimising the comparison statistics with snow cover MODIS data. In the specific application reported here, a value equal to 1.1 was used. Please note that since the corrected areal precipitation is an input to TOPMELT model, both PCF and SCF are not TOPMELT parameters.

*R2C2: Page 8, line 17: How does the conversion snow-water equivalent maps to snow cover maps work? What are the 'suitable threshold values'? I suggest the authors to integrate the description in the text.*

The comparison between model-based SWE and snow cover MODIS maps is carried out by converting the SWE map into a snow cover map using a threshold on SWE. In this paper, a minimum threshold of 10 mm was used. A reference (Parajka and Blöschl, 2008) and a brief note were added in section 4.3 of the revised paper. The revised paper is integrated with the following text: 'Then, the water equivalent maps may be converted to snow cover maps by using suitable threshold values (Parajka and Blöschl, 2008). In this work, we used a minimum threshold of 10 mm for the intercomparison with the MODIS snow cover products.'.

*R2C3: Page 9, line 9: The authors should report the estimated fractions of catchment area for the different land uses. Does the model consider interception by various vegetation covers? If so, how? How does the snowmelt module work in forested areas and are there any differences compared to other land uses?*

We thank the Reviewer for this note. The current version of the model does not consider land use for snow melt modelling. Accordingly, the model does not take into account

neither interception of snow by the vegetation and reduction of solar radiation during melting periods. A version of the model which includes a simplified canopy model is already available, but it is not included in this work, where we focus on the basic elements of TOPMELT (use of statistical distribution of solar radiation and assessment of the impact of its different simplified representations).

*R2C4: Page 13, lines 5-8: I think coniferous forests and discontinuous vegetation might be the dominant land covers in such a catchment. Therefore, given the limits of MODIS in vegetated areas, I suggest to show the comparison between TOPMELT and MODIS for all the land covers, except for the forested areas. As an alternative, the results of the comparison could be shown in a table or in the figures, but distinguishing the various land uses.*

We thank the Reviewer for this note. As we reported in our response to comment R2C3, land use and impact of vegetation is not considered in this work. However, we recognise the limits of MODIS in vegetated areas. To account for this, the revised version includes a new version of Figure 7 (now Figure 6 in the revised version), where we report the land use distribution. The revised version includes also text reporting the fractions of different land uses. This helps understanding the impact of forested areas on OA and better interpreting our results.

**Answers to minor comments**

*R2C5: Page 2, line 16: Please revise the English in this sentence.*

Revised (see marked-up manuscript).

*R2C6: Page 3, line 23: It is not clear why the fraction of glacier area or debris-covered glacier area is mentioned here without further explanation. Please integrate here with 1-2 brief sentences.*

We removed any reference to the debris covered glacier to avoid unnecessary information, given that this feature was not used in the case study. We revised the text

concerning the meaning of glacier area fraction.

*R2C7: Title 2.2: Does the model consider interception by different vegetation covers? And how?*

See R2C3 and R2C4.

*R2C8: Page 4, Line 11: It is not clear what interpolation technique is used in the model. Please explain whether there are different options in the model. And please give more details about the Precipitation Correction Factor. How is it obtained/computed? What are the typical values?*

See R2C1. The paper was revised to better explain how the precipitation input is processed and passed to the model (see marked-up manuscript).

*R2C9: Eq 2: Maybe capital letters should be used here.*

In the revised paper we use capital letters for parameters and lowercase for variables, except for temperature T, which would confuse with time t. We added a table with variables and parameter definition (see Table 1 below).

*R2C10: Page 8, line 18: Please explain and provide examples of these threshold values.*

See R2C2.

*R2C11: Page 9, line9: Given that glaciers cover a very small fraction of the catchment area, could you provide estimates for the remaining 93% of the area?*

See R2C4. Glaciers are included within the bare soil/rocks land use.

*R2C12: Page 10, line 18: I guess this is not the correct symbol.*

Thanks, corrected.

*R2C13: Page 10, line 28: Maybe you could consider reporting the figure as supplementary material or as the new Figure 4, if the current one will be merged with Figure*

*3.*

We merged Figures 3 and 4 in the revised version (see the following comment R2C14).

*R2C14: Fig 3-4: I think Figure 4 could be merged with Figure 3 in order to have only one figure but with two plots. Please consider this change.*

Done, see the new Figure 3 below.

*R2C15: Page 13, line 6: Please provide the fraction for the different land uses in the description of the study area. I think coniferous forests and discontinuous vegetation might be the dominant land covers in such a catchment. Therefore, given the limits of MODIS in vegetated areas, I would suggest to show the comparison between TOPMELT and MODIS for all the land covers, except for the forested areas. As an alternative, the results of the comparison could be shown in a table or in the figures, but distinguishing the various land covers.*

See R2C3 and R2C4. In the revised version (marked-up manuscript), we added text with the fraction of different land use cover.

*R2C16: Figure 5: Please report also in the caption of the figure what TP, TN, FP and FN represent.*

Done (see marked-up manuscript).

*R2C17: line2 Page 14: Please revise the English because the second part of the sentence is not clear.*

Done (see marked-up manuscript).

*R2C18: Fig 8: Please use the same scale for NSE and add a) and b) in the two plots and the caption of the figure.*

Done, see below revised Figure 7.

*R2C19: Page 15, line 11: Why should W4-C10 be the better choice considering ac-*

*curacy and computational efficiency? Based on Figure 8, I would choose W2-C10. Please explain the sentence (your choice) and especially from the computational efficiency point of view.*

For this case study, results obtained with the finest spatial and temporal aggregation are similar to those obtained with an intermediate configuration (W4-C10), despite W2-C10 performs better. TOPMELT, in its W4-C10 configuration, computes the snowpack model for 10 points (classes) and updates the radiation 12 times a year; on the other hand, the W1-C20 configuration performs double the load for computing the snowpack (20 classes) and updates the radiation distribution four times more (48 times a year). Therefore, an intermediate resolution represents a balance between computational efficiency and accuracy. We added this to the revised paper.

Please also note the supplement to this comment:
https://www.geosci-model-dev-discuss.net/gmd-2018-202/gmd-2018-202-AC1-supplement.pdf
* * *
[Figure]

**Figure 6.** a) Land use distribution for the catchment and b) pixel based overall accuracy (OA) of the comparison between simulated and MODIS-derived snow cover maps, computed from January 1 to June 30, 2011 for a total of 50 MODIS snow cover maps.

**Fig. 1.** Figure 6, revised paper.

**Table 1.** Model parameters and variables: short name, description and measuring units. Parameters are written with capital letters, variables in lowercase.

| Parameter | Description | Value | Units |
|---|---|---|---|
| $ALBG$ | Glacier albedo | 0.3 | - |
| $ALBS$ | Fresh snow albedo | 0.9 | - |
| $\beta_2$ | Dimensionless parameter for $alb$ computation | 0.0919 | - |
| $CMF$ | Combined Melt Factor | 0.013 | mm $^\circ$C$^{-1}$MJ$^{-1}$m$^2$ |
| $DYTIME$ | Speed of water propagation through snowpack | 3 | $mh^{-1}$ |
| $G$ | Precipitation gradient | 0 | km$^{-1}$ |
| $LWT$ | Water holding capacity, fraction of w.e. | 0.1 | - |
| $NMF$ | Night Melt Factor | 0.16 | mm $^\circ$C$^{-1}$h$^{-1}$ |
| $REFRZ$ | Freezing factor | 0.03 | mm $^\circ$C$^{-1}$h$^{-1}$ |
| $RI$ | Radiation Index, mean daily energy | $1 \div 42$ | MJ m$^{-2}$h$^{-1}$ |
| $RMF$ | Rain Melt Factor | 0.3 | mm $^\circ$C$^{-1}$h$^{-1}$ |
| $T_b$ | Base temperature | 0.0 | $^\circ$C |
| $T_c$ | Snow/rain threshold temperature | 1.5 | $^\circ$C |
| $WETH$ | Water equivalent minimum threshold before ice-melt | 5 | mm |

| Variable | Description | | Units |
|---|---|---|---|
| $alb$ | Snow albedo (accounting for aging) | | - |
| $h$ | Elevation | | m |
| $f$ | Fusion | | mm h$^{-1}$ |
| $ice$ | Freezed water | | mm |
| $liqw$ | Interstitial melt water | | mm |
| $p$ | Precipitation | | mm h$^{-1}$ |
| $T$ | Temperature | | $^\circ$C |
| $we$ | Water Equivalent (w.e.) | | mm |

**Fig. 2.** Table 1, revised paper.

[Figure]

**Figure 3.** a) Band migration index for the five temporal aggregations, reported for four elevation bands (lowest elevation band from 817 to 1000 m); from 1600 to 1800 m; from 2400 to 2600 m; highest elevation band from 3400 to 3485 m). b) Fraction of migrated pixels computed for the five temporal aggregations over all the elevation bands. Circles refers to pixels that migrated of ±1 energy class, squares to total migrated pixels.

**Fig. 3.** Figure 3, revised paper.

[Figure]

Figure 7. Box plots of NSE computed from the pixel by pixel comparison of the a) temporal and b) spatial aggregation series of w.e. maps, from October 2010 to June 30 2011 at weekly time-step. On each box, the central mark indicates the median, and the bottom and top edges of the box indicate the 25th and 75th percentiles, respectively. The whiskers extend to the maximum and the minimum efficiency. The reference is TOPMELT with W1-C10 configuration for the plot of the left panel, W4-C20 for the plot on the right.

**Fig. 4.** Figure 7, revised paper.

---

## Author Comment (AC2) · 9 Jul 2019

Our point-to-point response is reported below. Reviewer's Comments are reproduced in italics; the Authors' Responses are given directly afterward. All reviewer comments are identified using the code RXCY, where X is the reviewer number and Y is the reviewer comment number (for example R1C3 means Reviewer 1 Comment 3). Line numbers in authors' responses refer to the original manuscript unless otherwise stated.

Additionally, we enclosed a marked-up version of the revised manuscript.

**Answers to main comments**

[Figure]

*R3C1: The structure could be improved. 1) in the Introduction, the manuscript should emphasize the gap and importance of the research and how you are going to achieve, instead of model introduction as in lines 21-35 (page 2) and lines 1-10 (page 3).2) Section 4 kind of mixes the methodology and model results.*

In the revised version, we improved the Introduction by pointing out three important implications of integrating ETI snowpack models into semi-distributed runoff models. 'Integration of ETI snowpack models into lumped or semi-distributed hydrological models may have the potential to increase spatial transferability of calibrated snowpack model parameters for hydrological applications over ungauged mountainous basins, as shown by Comola et al. (2015). Another important implication of the stronger physical basis of the ETI model with respect to simpler 'degree day' models is that it might be more appropriate for the study of climate-change impact on melt regimes, as shown by Pellicciotti et al. (2005). Finally, increasing the accuracy of the modelled snow water equivalent may improve the outcomes of data assimilation procedures of remotely sensed snow cover information.' We substantially modified Section 4 by including analyses of scale dependency of radiation aggregation level. For this, we examined the control exerted by the catchment size on runoff simulations. We subdivided the study basin into a number of sub-basins characterised by different drainage areas. We isolated 5 basins with mean drainage of 20 km$^2$, 10 basins with mean drainage area of 10 km$^2$, and 20 basins with mean drainage area of 5 km$^2$. Results are reported in the new Section 3.4 of the revised version (also, see the answer to the next comment).

*R3C2: The model calibration is too short. For new model introduction, people are interested in model parameters, their sensitivities, or how to estimate these parameters. It is not clear here.*

The revised Sections 4.3 and 4.4 have been substantially modified, by including new work on the model sensitivity to spatial and temporal aggregation levels. We added a table (see Table 2 below) reporting the results of the validation for different model aggregation in space and time. It is evident from results that the model is not sensitive

to different configurations: this is due to the relatively large size of the basin. This basin size was chosen not to get best model performance, but to demonstrate model functionality in terms of output products and possible applications. As aforementioned in the answer to comment R3C1, we also examined the influence of the catchment size on runoff simulations. We subdivided the study basin into a number of sub-basins with different drainage areas. We isolated 5 basins with mean drainage of 20 km$^2$, 10 basins with mean drainage area of 10 km$^2$, and 20 basins with mean drainage area of 5 km$^2$. Results are reported in the new Section 4.4 of the revised paper. We added an additional table (Table 3, reported below) showing the sensitivity of the TOPMELT-ICHYMOD model to the catchment size .

*R3C3: Comparison results are also missing. For example, there is only one simulation presented in Figure 6, while there are so many simulations have been done (if I am right). How does snow parameters affect hydrograph or snow cover? (see also point 2 above).*

See our response to comment R3C2.

*R3C4: I am not clear how radiation is calculated. Is it based on station data or theoretical solar radiation equation? How did MODIS data come into play? Section 2.1 and/or section 4.1 could explain something on this.*

Radiation is computed theoretically based on the models mentioned in Section 2.1 'Clear sky potential radiation computation and derivation of radiation distribution'.

*R3C5: The integration of TOPMELT and ICHYMOD is also not clear, especially on the routing. My understanding is that ICHYMOD is a lumped model and its routing scheme shouldn't consider elevation bands. Then how is water from each cell (combination of elevation bands and radiation classes) routed to the outlet?*

ICHYMOD is a semi-distributed model which spatially aggregates the TOPMELT water output generated by the combination of elevation bands and radiation classes to

provide a lumped input to the model soil module (which accounts for infiltration and groundwater storage). Hence, the routing scheme is a lumped description of the water transfer at the basin scale, through both slow and fast pathways. We modified the text in the presentation of ICHYMOD to highlight this feature.

*R3C6: English is readable. However a native speaker might improve the manuscript*

Thanks, the work was checked by a native English speaker.

*R3C7: Section 5 seems more like a summary.*

We modified the conclusions to highlight the implications of the results obtained in this work (see the attaches marked-up manuscript).

**Minor comments**

*R3C8: Line 24 of page 1 and Line 1 of page 2. I don't understand the logic*

We are here introducing two extreme modelling approaches: the temperature index based modelling (simplified but efficient) and physical modelling (more realistic but complex). We do this to introduce TOPMELT approach, which is intermediate.

*R3C9: There is a duplication (line 29 of page 3, and line 1 of page 4)*

Corrected (see marked-up manuscript).

*R3C10: Equation 1.What is the range of G*

The range of G is limited to positive values. Since the precipitation gradient is linear with the elevation with slope governed by G, the precipitation can become negative at lower elevation bands for increasing values of G. In this case the code automatically limits the gradient, providing an automatic correction.

*R3C11: Figure 6: add the content of bottom plot in the caption.*

Content added (see marked-up manuscipt).

*R3C12: Figure 7: why don't you use the whole simulation period?*

We do not show the whole simulation period because we wanted to focus on an active snow melting phase. We chose this period in particular because the sample MODIS map of Figure 7 falls within this time range.

*R3C13: 'reference fields' in the second last line of Page 14: what are they?*

We changed 'fields' to 'w.e. distribution over space'. We better explained what we meant by 'reference' for Figure 8, both in the text and the caption (see the marked-up manuscript).

*R3C14: Figure 8: What is your point? To me, models with similar spatial or temporal resolution should give similar results.*

We wanted to yield In Figure 8 the sensitivity of modelled w.e. spatial distribution to different model configurations. To do so, the finest temporal and spatial model configuration was chosen as reference or Figure 8a and Figure 8b respectively. Additionally, please see our response to comment R1C4.

[revised manuscript text omitted]

---

## Author Comment (AC4) · 9 Jul 2019

Our point-to-point response is reported below. Reviewer's Comments are reproduced in italics; the Authors' Responses are given directly afterward. All reviewer comments are identified using the code RXCY, where X is the reviewer number and Y is the reviewer comment number (for example R1C3 means Reviewer 1 Comment 3). Line numbers in authors' responses refer to the original manuscript unless otherwise stated.

The following answers are an integration to the answers given with comment **SC1**. Re-organisation of the answers was necessary after a revision of the manuscript accounting for all the reviewers' comments. We apologise for any repetition.

[Figure]

**Answers to main comments**

*R1C1: LUMPED or SEMI DISTRIBUTED. Here is the definition of lumped very broad and actually the implementation with elevation bands and radiation index classes heavily reminds me the definition of hydrological response units, also a semi-distributed approach. I think your approach is much more semi-distributed than lumped.*

In the manuscript, we characterised TOPMELT as 'lumped' in order to make very clear the differences with a spatially distributed approach. However, we agree with the Reviewer that the model can be considered as 'semi-distributed', as this class of models do not make calculations for every point in the catchment but for a distribution function of characteristics. TOPMELT has the feature that the snow predictions can be mapped back into space for comparison with any observations of the snow properties. We substituted 'lumped' with 'semi-distributed' in the revised version.

*R1C2: INPUT PRECIPITATION Please expand on the techniques declared at page 4.*

With the sentence at line 13, page 4, we mean that precipitation can be calculated by using a range of methods (Thiessen Polygons, multi-quadratic and Kriging), on the condition that the model input is an areal precipitation estimated starting from any of these techniques. The interpolation and averaging code of precipitation is not included in the version of TOPMELT illustrated in this manuscript, but it is included in the complete hydrological model code. For the case study of the paper, we used the Thiessen polygons to calculate an areal precipitation over the whole basin. We clarified this issue in the revised version of the paper.

*R1C3: LIST OF VARIABLES I would welcome a Table with a list of the used abbreviations.*

A table with model parameters and variables is reported here below and is included in the revised version of the paper.

*R1C4: "DYNAMIC" RADIATION AREA AND INDEX: If you had static radiation regions*

[Figure]

*instead of radiation classes you would not need the supplementary workaround for updating the states with a "migration". Can you better justify your choice, or, even better, compare your results to a version with static radiation sub areas selected using elevation, aspect and/or slope?*

TOPMELT accounts for the seasonality of sun declination and for the visible horizon, therefore including both the effects of the temporal variability of the incident radiation angle and of shadowing. This makes the spatial distribution of radiation variable over time, which requires the updating of the snow states and represents a key feature of the model. The use of topographic variables, like aspect, elevation and slope as a surrogate for radiation is similarly subject to arbitrariness and lack of generality. For instance, during winter months in Norther Hemisphere, portions of north-facing slopes may remain shaded throughout the day due to the low angle of the sun. This causes snow on north-facing slopes to melt slower than on south-facing ones. The scenario is just the opposite for slopes in the Southern Hemisphere, where north-facing slopes receive more sunlight and are consequently warmer. Near the Equator, north- and south-facing slopes receive roughly the same amount of sunlight because the sun is almost directly overhead. At the Poles, north and south slopes tend to be either shrouded in darkness all winter long, or bathed in sunlight all summer long, with only slight variation between the slopes in spring and fall. All this, shows that using radiation instead of topographic variables leads to a better generalization of model application and evaluation. The paper already includes an analysis of the impact of decreasing the frequency of the snow state updating. This can be seen in Figure 8 of the submitted paper, left panel, which shows the impact of updating the radiation distribution at decreasing frequency. The frequency ranges from 1 week, which is chosen as the reference temporal aggregation, to 2 weeks, 4 weeks, 8 weeks and 12 weeks. Figure 1 below shows the scatter plots corresponding to the pixel-by-pixel comparison summarised in Figure 8 of the submitted paper, in terms of snow water equivalent (w.e.). The scatter reported in Figure 1 indicates that the impact of the decreasing frequency may have important consequences when the w.e. spatial distribution is sought.

**Specific comments**

*R1C5: Page 1, line 10: Similar topic was addressed in Klok et al, 2001, where Hock model has been implemented in a fully distributed and in a semi distributed model. Klok, L., K. Jasper, K. Roelofsma, A. Badoux, J. Gurtz (2001) Distributed hydrological modelling of a glaciated Alpine river basin. Hydrol. Sci. J. 46: 553-570.*

Reference added (see marked-up manuscript).

*R1C6: Page 1, line23: Zappa M, Pos F, Strasser U, Warmerdam P, Gurtz J. 2003. Seasonal water balance of an Alpine catchment as evaluated by different methods for spatially distributed snowmelt modelling. Nordic Hydrology 34: 179-202.*

Reference added (see marked-up manuscript).

*R1C7: Page 2, line 24: Here is the definition of lumped very broad and actually the implementation with elevation bands and radiation index classes heavily reminds me the definition of hydrological response units, also a semi-distributed approach.*

Please see our response to R1C1.

*R1C8: Page 2, line 28: A priori statement, not yet supported by results and/or references. The comment refers to the sentence: 'This is a potentially significant advantage when parameter sensitivity and uncertainty estimation procedures are carried out'.*

We have rephrased as follows: 'This is a potentially significant advantage when several model simulation runs should be carried out, such as in Monte Carlo based parameter sensitivity and uncertainty estimation procedures.'

*R1C9: Page 3, line 3: Making this a semi-distributed approach ....*

We agree on this comment, which underlines the need to term 'semi-distributed' the TOPMELT modelling approach.

R1C10: Page 4, line 8: single for the whole basin and a specific day or single for the

whole computation period?

We thank the Reviewer for the opportunity to better specify here: 'Air temperature data are used to estimate an unique hourly vertical lapse rate for the whole basin' (revised Section 2.2, first paragraph).

*R1C11: Page 4, line 13: Reference(s)? This is the only place where you declare how P is interpolated, but it seems to me quite strange to declare a "range of techniques" used .... Did you use now Kriging or Thiessen?*

See our response to comment R1C2. We modified the original text as follows: 'The model permits use of several techniques ranging from Thiessen's polygons to multiquadratic (Borga and Vizzaccaro, 1997) for the estimation of basin mean areal precipitation values. For the analyses reported in this work, the Thiessen method was used' (revised Section 2.2, first paragraph).

*R1C12: Page 5, line 8: Rerference(s)?*

Reference was added (Anderson, 1976).

*R1C12: Page 5, line 10: A table provided as supplementary material listing all variable and units might be a good addon.*

Please see our response to R1C3

*R1C13: Figure 1: You define for each time and elevation band 10 sub-regions with equal area after sorting them according to RI. Why 10 areas? Why not discriminate them according to slope and aspect (which seems dominant to me).*

'Why 10 areas?' The impact of using different subdivisions is examined in Figure 8b of the submitted paper, where the number of classes ranges from 1 to 20, showing that the gain in reproducing the snow water equivalent spatial distribution is very limited when more than ten classes are used. See our response to R1C4 for the comment concerning the use of topographic information to discriminate between local areas. We

revised Figure 1 changing the colour scale (see below).

*R1C14: Page 6, line 14: If you had static radiation regions instead of radiation classes you would not need this supplementary workaround for updating the states. Can you better justify your choice, or, even better, compare you results to a version with static radiation sub areas selected using aspect and/or slope?*

See our response to R1C4 for the comment concerning the use of topographic information to discriminate between local areas.

*R1C15: Page 8, line 17: What is suitable in your opinion?*

We used 10 mm as a threshold value in this analysis. The revised version has been updated accordingly and a reference supporting the choice was added (Parajka and Blöschl, 2008).

*R1C16: Page 10, line 3: Thanks, this replies one of my previous points.*

Thanks.

R1C17: Page 10, line 20: For nc1 there should not be any migration, isn't?

Yes, when using just one class there is no need to update the snow water equivalent.

*R1C18: Page 10, line 20: With static radiation classes you should not have any migration but exploiting the potential of ERI, isn't?*

The Reviewer is right in this remark, but we note that solar radiation is inherently variable in time. Thus, taking this variability into account should at least be attempted in a model which aims to use both temperature and radiation for snowmelt modelling.

*R1C19: Page 12, line 20: Zappa M. 2008. Objective quantitative spatial verification of distributed snow cover simulations – an experiment for entire Switzerland. Hydrological Sciences Journal, 53(1): 179–191. DOI: 10.1623/hysj.53.1.179.*

We added this reference.

*R1C20: Page 13, line 2: Still W4-C10?*

Yes, we put a note on this in the revised text (see marked-up manuscript).

*R1C21: Figure 6: I would be interested to see a "spaghetti plot" sorted by C and W.*

Whereas the sensitivity of the modelled snow water equivalent to variation of number of classes (C) and number of updates (W) is quite remarkable, the sensitivity of the modelled runoff is much less (actually, dispersion in the spaghetti plot cannot be recognised). This is due to the size of the study basin and the branching nature of the river network; both provide a powerful way in averaging out the heterogeneity of snowmelt processes, as shown by Comola et al. (2015). In order to illustrate this point, we reported values of the Nash-Sutcliffe index for different model simulations obtained by using different values of C and W (see Table 2 of the revised paper, reported below). In the revised version, we also examined the control exerted by the catchment size on runoff simulations. We subdivided the study basin into a number of sub-basins characterised by different drainage areas. We isolated 5 basins with mean drainage of 20 $km^2$, 10 basins with mean drainage area of 10 $km^2$, and 20 basins with mean drainage area of 5 $km^2$. Results are reported in the new Section 3.4 of the revised version (see Table 3, reported below, for a summary).

*R1C22: Figure 8: Why not using same scale of y-axis in the left and right graphs? So you could easily see that W is less sensitive than C*

In the revised version of the paper we used the same scale.

*R1C23: Page 15, line 5 (it is fig 8).*

Corrected.

**References**

Borga M. and Vizzaccaro A.: On the interpolation of hydrologic variables: formal equivalence of multiquadratic surface fitting and kriging, J. Hydrol., vol. 195, no. 1–4, pp.

[Figure]

160–171, doi:10.1016/S0022-1694(96)03250-7, 1997.

Comola, F., B. Schaefli, P. Da Ronco, G. Botter, M. Bavay, A. Rinaldo, and M. Lehning: Scale-dependent effects of solar radiation patterns on the snow-dominated hydrologic response, Geophys. Res. Lett., 42, 3895–3902, doi:10.1002/2015, 2015.

Parajka, J., and Blöschl, G.: Spatio-temporal combination of MODIS images - potential for snow cover mapping, Water Resour. Res., 44, W03406, doi:10.1029/2007WR006204, 2008.
[Figure]

[Figure]

Fig. 1: Scatter plot of the pixel-by-pixel comparison of w.e., obtained by updating the w.e. classes at decreasing frequency ranging from 2 weeks to 12 weeks. The updating frequency of 1 week is used as reference. The study period is from October 1 2010 to June 30 2011.

**Fig. 1.** Additional figure

**Table 1.** Model parameters and variables: short name, description and measuring units. Parameters are written with capital letters, variables in lowercase.

| Parameter | Description | Value | Units |
|---|---|---|---|
| $ALBG$ | Glacier albedo | 0.3 | - |
| $ALBS$ | Fresh snow albedo | 0.9 | - |
| $\beta_2$ | Dimensionless parameter for $alb$ computation | 0.0919 | - |
| $CMF$ | Combined Melt Factor | 0.013 | mm °C$^{-1}$MJ$^{-1}$m$^2$ |
| $DYTIME$ | Speed of water propagation through snowpack | 3 | $mh^{-1}$ |
| $G$ | Precipitation gradient | 0 | km$^{-1}$ |
| $LWT$ | Water holding capacity, fraction of w.e. | 0.1 | - |
| $NMF$ | Night Melt Factor | 0.16 | mm °C$^{-1}$h$^{-1}$ |
| $REFRZ$ | Freezing factor | 0.03 | mm °C$^{-1}$h$^{-1}$ |
| $RI$ | Radiation Index, mean daily energy | $1 \div 42$ | MJ m$^{-2}$h$^{-1}$ |
| $RMF$ | Rain Melt Factor | 0.3 | mm °C$^{-1}$h$^{-1}$ |
| $T_b$ | Base temperature | 0.0 | °C |
| $T_c$ | Snow/rain threshold temperature | 1.5 | °C |
| $WETH$ | Water equivalent minimum threshold before ice-melt | 5 | mm |

| Variable | Description | | Units |
|---|---|---|---|
| $alb$ | Snow albedo (accounting for aging) | | - |
| $h$ | Elevation | | m |
| $f$ | Fusion | | mm h$^{-1}$ |
| $ice$ | Freezed water | | mm |
| $liqw$ | Interstitial melt water | | mm |
| $p$ | Precipitation | | mm h$^{-1}$ |
| $T$ | Temperature | | °C |
| $we$ | Water Equivalent (w.e.) | | mm |

**Fig. 2.** Table 1, revised paper.

[Figure]

**Figure 1.** Comparison between radiation index distribution over the 2000-2200 m elevation band of the Aurino basin for a) January 1st and b) April 1st (ten classes subdivision). The figures show the north-eastern portion of the basin and report the average radiation index [J m$^{-2}$], with the corresponding radiation class identified by a roman number.

**Fig. 3.** Figure 1, revised paper.

**Table 2.** Nash-Sutcliffe index ($NSE$) of the TOPMELT-ICHYMOD model at different spatial aggregation and temporal resolution, from October 2001 to October 2007.

| W4C1 | W4C5 | W4C10 | W4C15 | W4C20 |
|---|---|---|---|---|
| 0.73 | 0.73 | 0.71 | 0.73 | 0.73 |

| W1C10 | W2C10 | W4C10 | W8C10 | W12C10 |
|---|---|---|---|---|
| 0.71 | 0.71 | 0.71 | 0.70 | 0.71 |

**Fig. 4.** Table 2, revised paper.

**Table 3.** Mean value of the Nash-Sutcliffe index (NSE) of the comparison between W4C1 and W4C10 TOPMELT-ICHYMOD simulated flows and the reference flow simulations, obtained by using the W4C20 set up, over basins of three different drainage areas: 5, 10 and 20 km$^2$. Comparisons carried out over the March, 1 to June, 30 period.

| Model set-up | Sub-basin area | | |
|---|---|---|---|
| | 5 km$^2$ | 10 km$^2$ | 20 km$^2$ |
| W4C1 | 0.77 | 0.91 | 0.99 |
| W4C10 | 0.97 | 0.99 | 0.99 |

**Fig. 5.** Table 3, revised paper

---

## Referee Report (RR1)

[referee-annotated manuscript omitted]

---

## Author Response (AR2)

Thanks again to the Editor and the three Reviewers for the efforts spent on helping us driving our manuscript to its current version. We accepted all the corrections from Reviewer 1, as indicated through the reported manuscript and listed in the following section.

Also, we report questions (in blue) and relative answers to Reviewer 2, whose point of view we integrally share. Corrections made on the revised manuscript are described within the answers.

**Review 1**

Page 1 lines 7, 10 and 12. Corrected.

Page 10 lines 21 and 22. Corrected.

Page 11 lines 1 and 5. Corrected.

Page 11 lines 3 and 30. Corrected.

Page 11 line 4. Corrected.

Page 17 line 4. Corrected.

Page 19 lines 22 and 24. Corrected.

**Review 2**

**Main comments:**

1. That is good to list all model parameters in Table 1 and also good that you studied the sensitivities of temporal and spatial resolutions on model results. I wonder if this sensitivity study can be applied on those parameters – and possibly the meaning of these calibrated parameters in the application.

The most important TOPMELT parameters are the albedo, the combined melt factor, the precipitation gradient, the base temperature and the snow/rain threshold temperature. Additional work is ongoing to assess model sensitivity to snow parameters, assessing TOPMELT on other models, snow ground records and satellite observations of the snow-covered area.

For completeness, we added a brief description of the main snowpack parameters.

2. I wonder if authors have compared this ETI method with other snowmelt models, or even the ICHYMOD with snow module. This might promote the importance of snowmelt and your model a lot.

A work with interesting outcomes on TOPMELT comparison with other distributed models is ongoing.

3. English is readable. However, it could be improved. E.g. Lines 16-18 in Page 12 can be rewritten as: It is worth noting that the migration index increases with the number of radiation classes (results not shown here for brevity).

Thanks, the sentence was rewritten.

**Other comments:**

1. Line 1 in Page 4: you wrote "one of the nb*nc model cells …". Does this mean only one computation unit is allowed for glacier? If this true, my understanding is that the elevation bands are conceptually divided into cells instead of spatially. Does this mean RI is also conceptually given?

Thanks for the comment, we meant "each one of the nb*nc model cells". Of course, variables are evaluated for each single radiation class within each single elevation band.

2. In section 2.3, there is a difference in day melt and night melt. However, in your example, the finest temporal resolution is 1 week. Can you give a bit more explanation?

TOPMELT computes snowpack processes at hourly time resolution. When we say that the temporal resolution is one week, we mean that radiation is computed at hourly temporal resolution, aggregated over one week, and then applied at hourly temporal resolution. Additionally, to aim the best computational performance, computation of radiation is performed once for all by a pre-processing code and embedded in the structure of each application: in this case the Aurino catchment for which radiation was computed on the basis of a 30 m resolution DTM.

Computation of radiation yields the spatial model domain, which can be variable at any temporal resolution, from hours to months. Regarding night and day snowmelt, the first is computed regardless the radiation; instead, the second is calculated by means of the radiation index, which is an input parameter and is pre-computed as mentioned above.

3. Table 1: check the value for "RI"

Thank you. We realized that some confusion may arise when working with power and energy. We take advantage of this opportunity to definitely amend the definition of this index. The Radiation Index is indeed a daily energy and its measuring unit is $MJ/m^2$, not $MJ/m^2/ h^{-1}$, which stands for mean hourly power. We corrected the measuring unit of RI and of the Combined Melt Factor (CMF) in Table 1. We also corrected the same units after Equation 2 and how the radiation index is implemented in the same equation.

4. Line 3 in Page 10: what is "net precipitation"

It is the excess liquid precipitation plus melt contribution (from glacier and snowpack). "Net precipitation" was changed to "excess precipitation plus melt contribution".

5. Line 1 in Page 11: you used a fixed number of elevation bands. Is there any consideration for this?

TOPMELT can implement any number of elevation bands. The sensitivity to the vertical subdivision is an additional issue deserving some attention, for example when dealing with problems such as the seasonal evolution of the snow-covered area. We did not investigate this issue in this paper because we were mainly interested to the spatial and temporal resolution effects on snow water equivalent distribution among classes within the elevation bands.

[revised manuscript text omitted]